# Propranolol: A “Pick and Roll” Team Player in Benign Tumors and Cancer Therapies

**DOI:** 10.3390/jcm11154539

**Published:** 2022-08-04

**Authors:** Virginia Albiñana, Eunate Gallardo-Vara, Juan Casado-Vela, Lucía Recio-Poveda, Luisa María Botella, Angel M Cuesta

**Affiliations:** 1Centro de Investigaciones Biológicas Margaritas Salas, 28040 Madrid, Spain; 2CIBERER, Centro de Investigación Biomédica en Red de Enfermedades Raras, Unidad 707, Instituto de Salud Carlos III (ISCIII), 28029 Madrid, Spain; 3Yale Cardiovascular Research Center, Section of Cardiovascular Medicine, Department of Internal Medicine, Yale University School of Medicine, 300 George Street, New Haven, CT 06511, USA; 4Departamento Bioingeniería, Escuela Politécnica Superior, Universidad Carlos III de Madrid, Av. de la Universidad, 30, 28911 Leganés, Spain; 5Facultad de Ciencias Experimentales, Universidad Francisco de Vitoria, 28223 Pozuelo de Alarcón, Spain; 6Departamento de Bioquímica y Biología Molecular, Facultad de Farmacia, Universidad Complutense de Madrid, 28040 Madrid, Spain

**Keywords:** beta-adrenergic receptor antagonist, propranolol, chemotherapy, combination cancer therapy, HIF, apoptosis, inflammation, angiogenesis, biomarker

## Abstract

Research on cancer therapies focuses on processes such as angiogenesis, cell signaling, stemness, metastasis, and drug resistance and inflammation, all of which are influenced by the cellular and molecular microenvironment of the tumor. Different strategies, such as antibodies, small chemicals, hormones, cytokines, and, recently, gene editing techniques, have been tested to reduce the malignancy and generate a harmful microenvironment for the tumor. Few therapeutic agents have shown benefits when administered alone, but a few more have demonstrated clear improvement when administered in combination with other therapeutic molecules. In 2008 (and for the first time in the clinic), the therapeutic benefits of the β-adrenergic receptor antagonist, propranolol, were described in benign tumors, such as infantile hemangioma. Propranolol, initially prescribed for high blood pressure, irregular heart rate, essential tremor, and anxiety, has shown, in the last decade, increasing evidence of its antitumoral properties in more than a dozen different types of cancer. Moreover, the use of propranolol in combination therapies with other drugs has shown synergistic antitumor effects. This review highlights the clinical trials in which propranolol is taking part as adjuvant therapy at single administration or in combinatorial human trials, arising as a good pick and roll partner in anticancer strategies.

## 1. Introduction

The different cancer radio- and chemotherapies have always been inspired by Paul Erlich’s *Zauberkugel*. Consequently, as long as new key molecules on different tumoral processes such as angiogenesis, cellular signaling, stemness, metastasis, and drug resistance and inflammation appeared, new specific single reagents and strategies rose up. We can summarize them all and make a starting lineup composed of alkylating agents, anti-metabolites, anti-tumor antibiotics, mitotic and topoisomerase inhibitors, growth factor inhibitors, corticosteroids, and immunological reagents. Nevertheless, despite the initial promising results on primary tumor shrinkage, non-desired effects, such as tumor resistance, relapse, metastases, and cytotoxicity, have been reported [1]. In consequence, many have been excluded from the first line of anticancer treatments, and a number of different new strategies attempt to overcome these drawbacks, such as personalized, combinatorial, or drug repurposing.

Adrenergic receptors (ARs) are a class of G-protein-coupled transmembrane receptors (GPCRs) that promote cellular proliferation and survival [2] when activated by small ligands such as the catecholamines, adrenaline and noradrenaline [3]. Unfortunately, overstimulation or signaling dysregulation of GPCRs frequently leads to cancer disease because of the alteration of processes such as survival and proliferation, immune system evasion, tumor angiogenesis, invasion, metastases, and drug resistance [4] (Figure 1A).

Propranolol (C_16_H_21_NO_2_), as shown in Figure 1B, is a relatively small molecule (259.349 g·mol^−1^) including a secondary amine group (PubChem CID: 4946). Propranolol is a nonspecific antagonist of β1 and β2 adrenergic receptors (ADRB1-2). Initially indicated for cardiac disorders, multiple clinical therapeutic properties outside of its first indication have been discovered throughout its nearly 60 years on the market. These properties include the ability to cross the blood–brain barrier (BBB) vasoconstriction, inhibition of angiogenesis, and induction of apoptosis [5,6]. In addition, accumulated publications and clinical trials point to its possible role as a regulator of the tumor microenvironment (see Section 2. “Why propranolol?”). As a paradigmatic example, propranolol has become the most valuable player drug in the benign tumor infantile hemangioma (IH) (see Appendix A).

Furthermore, the “*starting line up*” targets in cancer therapies have been: (1) the cell cycle (proliferative signaling, cell death, and replicative immortality), (2) cellular metabolism (the Warburg effect), (3) the tumor microenvironment (immune response, inflammation and, angiogenesis), and (4) multidrug resistance. We propose propranolol as a good “*fifth team player* for *pick and roll”* strategies in anticancer combination therapies. This review highlights the therapeutic benefits of propranolol (alone or in combination) described on a broad landscape of cancer treatments in human trials, including benign and rare carcinomas.

## 2. Why Propranolol?

More than a century ago, scientists speculated that catecholamines, after binding selectively to receptor-like structures, caused their pharmacological actions. It was in 1948 when R. P. Ahlquist described, for the first time, the α-and β-adrenotropic receptors [7]. Some years later, Sir James Whyte Black, a Scottish pharmacologist at Imperial Chemical Industries (ICI) in the UK, was searching for a chemically synthesized agent to interfere with catecholamines and, therefore, decrease the oxygen heart requirements and relieve the pain of angina. He was searching for a β-adrenergic blocker.

There were two main candidates: pronethalol and propranolol (also named ICI 45,520 or, since 1964, marketed as Inderal). The latter is a more potent derivative from the former and lacks the carcinogenicity found in animal models treated with pronethalol. Then, propranolol became a best-selling drug used to treat a wide range of cardiovascular diseases, such as arrhythmia, hypertension, and hypertrophic cardiomyopathy. Finally, in 1988, Sir James W. Black was awarded the Nobel Prize in Medicine for the creation of propranolol [8].

Nowadays, propranolol has fallen from the first line of treatment since it does not perform as well as other drugs. Nevertheless, evidence began to accumulate, paving the potential use (reposition) of propranolol as an antitumor drug. This use is supported by several case reports [9,10,11,12] that postulated the use of propranolol in a rare tumor-like pheochromocytoma to reduce hypertension generated by catecholamines’ secretion. Interestingly, these studies outlined (but did not relate) the absence of metastases and tumor growth amelioration without noticeable side effects, together with the normalization of the heart rate and blood pressure. A few years later, different reports started to show the clinical benefits of propranolol in different hyperproliferative diseases, such as astrocytomas [13], epidermoid carcinoma [14,15], and malignant insulinoma [16], lung adenocarcinoma [17], and breast cancer [18].

Nevertheless, propranolol continued on the “*sideline”* as cardioprotective medicine until 2008, when Léauté-Labrèze et al. showed its clinical antiangiogenic therapeutic properties in 11 cases of infantile hemangioma (IH) [19]. IH is a benign vascular tumor with a 4–5% of prevalence in the neonate and affecting mainly the face and limbs. Before the advent of propranolol, IHs surgery was the only treatment if IHs were not spontaneously remitting. After several case reports and successful trials, Hemangiol (an oral liquid presentation of propranolol) was designated as a drug for IH treatment by the European Medicines Agency (EMA) in 2014 [20,21,22].

Furthermore, several in vitro and in vivo studies described the antitumor properties (viability, oxidative stress, or metabolic and immunity restoration) of propranolol in different carcinomas, including breast [23], colon [24], gastric [25], glioblastoma [26], lung [17,27,28], nasopharyngeal [29], ovarian [30,31], and pancreatic [25,32,33,34].

Propranolol has properties as a membrane stabilizer and is able to cross the blood– brain barrier due to its lipophilic nature [6]. In relation to its anti-tumoral properties, three main mechanisms have been described: vasoconstriction, the induction of apoptosis, and the inhibition of angiogenesis (Figure 2).

### 2.1. Vasoconstriction

Adrenaline, via ADRB2, promotes vasodilation in pericytes and smooth muscle cells [35]. As a G-coupled protein receptor (GPCR), the α-subunit of the G-protein triggers adenylate cyclase (AC), converting ATP to cyclic adenosine monophosphate (cAMP). The second messenger, cAMP, activates cAMP-dependent protein kinase A (PKA), which will phosphorylate the proteins modulating their activity. Propranolol reverses the action of adrenaline, causing vasoconstriction, reducing blood flow in the vessels feeding the tumor, and slowing its growth [36].

### 2.2. Inhibition of Angiogenesis

To the best of our knowledge, the complete mechanism of action of propranolol has not yet been revealed, but some steps and facts related to its antiangiogenic properties are beginning to be elucidated. The first description of the consequences (apoptosis) of treating endothelial cells with propranolol was in 2002 [37], but the true antiangiogenic data began in 2009 by Annabi et al. [38]. Since then, in vitro assays with endothelial cells from different tissues, such as the umbilical cord (human umbilical vein endothelial cells, HUVEC) [39], brain (human brain microvascular endothelial cells, HBMEC), embryonic stem cells [40], hemangioma [41,42] and its stem cell derivative, [43,44,45] and hemangioblastoma from von Hippel–Lindau (VHL) disease [46,47], from neuroblastoma [48] and white blood cells (leukemic and monocytes) [49], and in vivo assays on retinopathies [50] and vascular lesions [51] have demonstrated the antiangiogenic effects of propranolol. The research developed on these cells demonstrate that propranolol exerts direct antiangiogenic effects by decreasing the expression levels of key molecules on the angiogenic process, such as VGEF [40,41,42,43,46,49,50] and its receptor VEGFR [39,40,41,42], HIF1-α [40,46,50], bFGF [43], FGF-2 [40], MMP-9 [38,42], MMP-2 [39,49], EPO [46], p-cofilin [42], vascular markers as CD31 and VE-cadherin [40], and vascular regulators, such as eNOS and NO [40]. Propranolol also interferes in signaling pathways which end in the impairment of the angiogenic process; some of them are the following: cell cycle arrest by the downregulation of cyclins A2, D1–3 and the upregulation of p15, p21, and p27 [39,42], and the inhibition of the pI3K/Akt [41,42], p38/MAP [41], and ERK1/2 signaling pathways [39].

### 2.3. Induction of Apoptosis

Propranolol blocks ADRBs, inducing apoptosis in different cell types and tumors in vitro, e.g., in ECs [37], pancreatic carcinoma cells [32], gastric carcinoma [25], and breast cancer [52]. We hypothesized that β-adrenergic antagonists counteract the Src-MAPK-mediated inhibition of apoptosis caused by agonists by increasing apoptosis.

More recently, our group has shown that propranolol induces apoptosis in HeLa, renal carcinoma, and hemangioblastoma (HB) primary tumor cell cultures from patients with von Hippel–Lindau disease (VHL) [53,54,55]. The induction of apoptosis, therefore, would represent another favorable mechanism of propranolol action in the treatment of tumor processes.

## 3. Propranolol as Adjuvant Therapy at Single Administration in Clinical Trials

### 3.1. Infantile Hemangioma

Since the above-mentioned breakthrough of propranolol for IH in 2008 [19], 22 different clinical trials (Appendix A) have been registered in the EU, the USA, Australia, and New Zealand. These trials were mainly conducted in newborns and toddlers; only one of them included children up to 5 years of age. The results invariably show the success of propranolol in decreasing or eliminating IH. In these trials, propranolol was used from 1 to 3 mg/Kg body weight/day with a length of the treatment between 3 to 12 months.

In phase 2/3 (NCT00744185), 14 patients were randomly assigned to the group receiving propranolol as monotherapy (seven patients) or placebo (seven patients). By the fourth week of treatment, IH growth had stopped in all infants in the propranolol group. There was a statistically significant difference between groups in the percentage change in IH thickness at week 4 [56]. The phase 2 trial ACTRN12611000004965 followed infants from 9 weeks to 5 years of age. Propranolol hydrochloride, administered orally at 2 mg/kg/day, reduced the volume, color, and focal and segmental elevation [57].

In 2010, Leaute Labreze initiated a phase 2/3 trial to demonstrate the efficacy and safety of systemic propranolol in infants with IH (NCT01056341). Four hundred-sixty patients completed 24 weeks of oral propranolol at 3 mg/kg/day. The data showed a higher frequency of treatment success with propranolol (60% vs. 4%, *p* < 0.001). Additionally, known adverse events associated with propranolol (hypoglycemia, hypotension, bradycardia, and bronchospasm) occurred occasionally, with no significant differences in their frequency between the groups [22].

A single-arm, phase 3-multinational study was conducted in patients aged from 35 to 150 days with high-risk IH (EudraCT Number: 2014-005555-80) [58]. The study comprised a 6-month initial treatment period (ITP) plus a continuation until 12 months of age. The patients received oral propranolol twice daily (3 mg/kg/day). The results showed a success rate of 47% at 6 months, which increased to 76% at the end of the ITP. Of the patients who achieved success, 68% continued for 3 months without treatment, and 24% required retreatment. Adverse events, reported by 80% of the patients, were mild and expected in this population related to known side effects of propranolol. The conclusions of this trial were that long-term propranolol administration significantly increased the success rate in high-risk IH. Success was sustained in most patients up to 3 months after stopping the treatment. Retreatment was effective, and the safety profile was satisfactory. Following these satisfactory results with relatively mild side effects, the EMA authorized, in 2014, the designation of Hemangiol as a new commercial presentation of propranolol (an oral liquid presentation of propranolol adapted for infants) for the treatment of IH.

A comparison between propranolol and previous reagents for IH has also been performed. The efficacy of prednisolone was compared with that of propranolol in two phase 2 trials, NCT00967226 [59] and NCT01908972 [60]. The results showed similar efficacy in reducing IH proliferation. While prednisolone showed a faster response rate, propranolol was better tolerated with significantly fewer severe adverse events, leading to the conclusion that propranolol should be the first line of IH unless propranolol was not tolerated (Appendix A).

Atenolol, a specific ADRB1-blocker, has also been analyzed in two phase 3 trials, NCT02342275 [61] and NCT03237637 [62]. The former was a prospective, multicenter, randomized, controlled, open-label clinical trial conducted in collaboration between six different research centers in China. A total of 377 patients were randomized to the propranolol (190) or atenolol (187) groups. Data analysis revealed that atenolol had similar efficacy to propranolol and fewer adverse events in the treatment of infants with problematic IHs, suggesting that oral atenolol could be used as an alternative systemic treatment for IH patients. However, the latter was initiated in 2017, and no results are yet available to corroborate or contradict the previous trial (Appendix A).

In the event of an emergency, a safe and rapid initiation protocol seems crucial. The latter is even more true in those cases in which a pediatric cardiology consultation is required. Therefore, a retrospective, longitudinal, descriptive, and observational phase 4 study was conducted, including 154 children under 2 years old and hospitalized for the introduction of Hemangiol (NCT04105517). The trial evaluated the safety, effectiveness, tolerance, and cardiological changes. The results obtained showed that no severe adverse events occurred in the rapid protocol and that there was good cardiac tolerance [63].

### 3.2. Propranolol for Solid Tumors

#### Preoperative Propranolol in Breast Cancer

Imitating the initial applications of propranolol on pheochromocytoma, it has also been used as a preoperative adjuvant in breast cancer. Metastases are the main cause of death in breast cancer patients, a process that may be inhibited by β-blockers, according to preclinical studies [64,65]. The randomized phase 2 study (ACTRN12615000889550) investigated whether, compared with a placebo, preoperative propranolol (7 days) modified its cancer gene expression. The conclusions were that one week of propranolol reduced intratumoral mesenchymal polarization and promoted immune cell infiltration in early-stage surgically resectable breast cancer. These results show that the ADRB-blockade reduces the biomarkers associated with metastatic potential, and they support the need for larger phase 3 clinical trials to detect the impact of the ADRB-blockade on cancer recurrence and survival.

In phase 2 NCT02596867, the aim was to evaluate the effect of propranolol on reducing the tumor proliferative index using Ki-67 as a biomarker, as well as the effect on safety, toxicity, and adherence to propranolol [66]. The results showed that propranolol decreased the expression of the pro-proliferative Ki-67 and the pro-survival Bcl-2 markers and increased pro-apoptotic p53 expression in a patient with stage III breast cancer. Molecular analysis revealed that the ADRB-antagonism disrupted the cell cycle progression and steady-state levels of cyclins. Furthermore, the propranolol treatment of breast cancer cells increased p53 levels, enhanced caspases cleavage, and induced apoptosis.

### 3.3. Propranolol for Non-Solid Tumors

#### Multiple Myeloma

A two-armed phase 2 trial was conducted to analyze the efficacy of propranolol in decreasing the gene expression of stress-mediated β-adrenergic effects in 25 individuals with multiple myeloma who received an autologous hematopoietic cell transplant (HCT) (NCT02420223) [67,68]. The preclinical research showed that the stress-induced activation of the sympathetic nervous system could promote hematopoietic malignancies through ADRB-mediated molecular pathways. HCT recipients exposed to chronic stress showed the activation of a conserved transcriptional response to adversity (CTRA) gene expression profile associated with disease-free survival. This randomized controlled phase 2 biomarker trial tested the impact of propranolol on CTRA-related gene expression. Propranolol was administered for 1 week prior to and 4 weeks after HCT, and propranolol-treated patients showed greater decreases from the baseline to HCT day −2 and day +28 for both CTRA genes. The changes in these pathways make propranolol a potential candidate for adjunctive therapy in cancer-related HCT (Figure 3).

### 3.4. Propranolol in Observational Studies

Since propranolol has not yet been approved for malignant tumors, observational studies where propranolol has been used are included in Appendix A. Among them, four were related to IH and one to pediatric cancers. NCT04105517 is a phase 4 Hemangiol postmarketing study conducted to look for adverse events in 500 children treated with propranolol for IH. Retrospective NCT04651049 collects parameters of the physiological development in children treated with propranolol due to IH. The only observational study including pediatric cancer is NCT02165683, where propranolol was used as an adjuvant with standard chemotherapies for different pediatric cancers. The aim of this study was to reduce the fluorodeoxyglucose (FDG)-positron emission tomography (PET) uptake as an indicator of improvement in cancer treatment. As seen in Appendix A, no results are available, although the studies have been completed.

## 4. Propranolol in Combinatorial Therapy

Early studies using propranolol in combination with other reagents, apart from the case reports already discussed in pheochromocytoma, sought to reduce preoperative stress in rats and, thereby, enhance the antitumor response of the chemotherapeutic agent (synergistic effect). The hypothesis was that secreted catecholamines and prostaglandins under stress conditions: (i) impair the host cell-mediated immunity (CMI) driven by NK cells, (ii) trigger inflammatory processes leading to angiogenic responses, and (iii) fuel tumor growth [69,70,71,72]. Therefore, a preoperational combination of an ADRB-blockade with propranolol, plus treatment with the COX-2 inhibitor, Celecoxib, would reduce lung metastases, enhance the antitumor activity of the NK cells, and increase the survival rates in different xenografts, as has already occurred [69,73,74].

Furthermore, new trials have been supporting the synergistic effect of propranolol as an adjuvant in combination therapies with cytotoxic 5-FU [23], TKI sunitinib [75], an anti-PD-1 [76] plus IL-2, the electron transport chain inhibitor, metformin [77], and the glycolysis inhibitor, 2DG [78].

All this has inspired an increasing number of clinical trials investigating the use of propranolol as an antiangiogenic drug in combination with some other anticancer drugs used as a plausible tumor therapy in pre- and post-operative patients. Some of these clinical trials described below appear to be promising combination therapies for different angiogenesis-related diseases, although most of them have either not been completed or have not yet shown conclusive results.

### 4.1. Propranolol and Etodolac (COX-2 Inhibitor)

Propranolol has been used in combination with Etodolac, a non-steroidal anti-inflammatory drug (NSAID) that acts as a competitive inhibitor of Cyclooxygenase 2 (COX-2), in four different studies as a preoperative combination therapy with propranolol for colorectal, pancreatic, and breast neoplasms.

#### 4.1.1. Breast Cancer

The results of the randomized, placebo-controlled, multicenter trial in women undergoing breast cancer surgery (NCT00502684), with 38 patients treated from pre-operative day 3 to post-operative day 2, showed a decrease in tumor cell migration capacity and subsequent metastasis by reducing the epithelial to mesenchymal transition (EMT), premetastatic/proinflammatory transcription factors, such as GATA-1 and the early-growth response-3/EGR3, tumor-infiltrating monocytes, and inflammatory cytokines, such as IL-6 [79,80,81] (Appendix A). The primary clinical endpoint was the 5-year recurrence rate, but no results related to this outcome have been published yet. Notably, no adverse effects have been observed so far.

#### 4.1.2. Colorectal Cancer

The multicentered, randomized, double-blind, placebo-controlled phase 3 trial, with 34 colorectal cancer patients treated for 20 consecutive days, starting 5 days before tumor extraction, showed a reduction in the stress-inflammatory response by a preoperatively combined therapy of propranolol and etodolac, and during colorectal cancer surgery, a decreased metastatic risk [82]. Two other clinical trials, NCT00888797 [83], a prospective multicenter randomized phase 3, and NTC03919461 [84], a double-blind placebo-controlled two-arm phase 2 with 400 and 200 colorectal cancer patients, respectively, are parallel trials to the one described above (Appendix A). Although these trials have not yet been completed and the results of these trials have not been published, the primary outcomes are local and distant, or a metastatic recurrence rate at three years and disease-free survival at three years, overall survival at five years, and biomarkers in the blood samples.

#### 4.1.3. Pancreatic Cancer

The results obtained in the above-mentioned trial (NCT00502684) on breast cancer [82,83] aimed to run a randomized, double-blind placebo-controlled two-arm interventional human trial (NCT03838029) [85] (Appendix A). This phase 2 trial is expected to have 210 pancreatic cancer patients undergoing curative surgery. Perioperative treatment with propranolol and etodolac will be performed for 35 days and initiated 5 days before surgery. The primary outcome measures are a cancer recurrence rate of up to 60 months after surgery, EMT, immune cell markers, and inflammatory cytokine levels in the blood samples for one year after surgery.

#### 4.1.4. Neuroblastoma

Neuroblastoma is one of the most frequent causes of cancer death in childhood, and there is an urgent need to find a suitable therapy with low toxicity. The trial NCT02641314 is a randomized phase 2 of metronomic chemotherapy in 26 patients aged 2 to 21 years old and newly diagnosed with high-risk neuroblastoma that progressed despite prior treatment (Appendix A). In this clinical trial, propranolol is used in combination with four drugs: the immunomodulating drug Celecoxib (which modulates the immune response by acting as an anti-inflammatory and inhibiting COX-2) and chemotherapeutic drugs, such as cyclophosphamide, etoposide, and vinblastine. During the trial, the doses and timing of each of these vary. The primary outcome is to demonstrate that this therapeutic combination could reduce the progression and/or recurrence from the start of the treatment up to 12 months. This phase 2 is based on a first study described by Berthold et al. (2017), in which they treated, with the same drug combination, except for propranolol, 20 patients with recurrent and refractory high-risk neuroblastoma [86]. They concluded that the secondary event-free and survival curves had no significant differences and that this metronomic approach is similarly effective as the standard treatment but with lower toxicity. These results allowed the development of the new ongoing clinical trial in which propranolol has been added as a fifth non-toxic antiangiogenic drug. The primary and secondary outcomes include a non-inferiority of event-free survival and a disease control rate at 6 months and overall survival at 12 months.

### 4.2. Propranolol and Prednisolone (Corticosteroid)

Among the clinical trials conducted in IH, there is one with the combined therapy of prednisolone (a standard corticosteroid treatment) and propranolol, NCT01074437 [87] (Appendix A). In this phase 2, with 9 patients enrolled, standard treatment with corticosteroid alone was compared with a placebo and combination therapy of propranolol and prednisolone. Although the idea was promising, no results were obtained because of insufficient enrollees to perform the randomization, which limited the number of participants and made the project unfeasible. At present, there are no ongoing combination trials of propranolol for HI.

### 4.3. Propranolol and Standard Chemotherapy

The following clinical trials have not published results or are active at the time of writing this review, as explained above. Although we cannot conclude about the therapeutical benefits, these trials point to the fact that propranolol is gaining weight in the field of cancer therapies.

#### 4.3.1. Hepatocellular Carcinoma (HCC)

The phase 2 trial NCT01265576 is active (Appendix A) and studies sorafenib, a tyrosine kinase inhibitor for patients with advanced HCC and with or without VT-122 (propranolol plus etodolac already mixed as a coadministration drug) [88]. The aim of this study is to assess whether the use of VT-122 is safe and effective in cachectic patients with advanced HCC and systemic inflammation and whether the treatment can improve both survival and quality of life. In the two experimental groups, sorafenib with placebo and sorafenib plus VT-122, all the participants in the sorafenib group will be treated at least 30 days prior to the randomization. The results are not reported at this time.

#### 4.3.2. Ovarian Cancer

NCT01504126 is an early phase 1 trial in patients with ovarian, primary peritoneal, or fallopian tube cancer treated with a combination of propranolol and standard chemotherapy (paclitaxel/carboplatin) [89] (Appendix A). Thirty-two participants (including stages II to IV of the above tumor types) are enrolled. Serum levels of IL-6, IL-8, and VEGF will be measured as changes in the immune response from the baseline (the treatment initiation and before the surgery) to 6 months in patients who successfully complete six cycles of chemotherapy in the absence of disease progression or toxicity. 

The combination of chemotherapy with propranolol resulted in improved quality of life, with particular emphasis on anxiety. A decrease in inflammatory biomarkers was also observed [90].

#### 4.3.3. Breast Cancer

NCT01847001 is a clinical trial involving 10 patients [91] (Appendix A). The patients received two types of chemotherapy regimens: paclitaxel or doxorubicin, and cyclophosphamide, both plus a propranolol treatment. After completion, they underwent surgery to remove the breast tumor. As adverse events, one patient suffered gastrointestinal disorders and neutropenia after the treatment and resection, and two of them suffered other cardiac disorders. The results support the combination feasibility of propranolol (up to 80 mg ER) with neoadjuvant taxane/anthracycline-based chemotherapy.

#### 4.3.4. Esophageal Adenocarcinoma

The ongoing parallel randomized phase 2 NCT04682158 combines propranolol with the standard chemoradiation of carboplatin and paclitaxel for esophageal adenocarcinoma on 60 patients (Appendix A). The primary outcome of this phase 2 is the survival and progression for up to 5 years, and it is actively recruiting patients [90].

#### 4.3.5. Sarcoma and Melanoma

The NCT03384836 single-group, phase 1b/2, studies the side effects, toxicity, and efficacy of propranolol administered together with pembrolizumab (the monoclonal antibody used to boost the anti-tumor immune response) (Appendix A). The treatment is planned for 47 patients with stage IIIC-IV melanomas that cannot be removed by surgery. The primary endpoint measures are adverse events, the treatment dose toxicity, and the overall response rate, analyzing the levels of some biomarkers from the baseline to 2 years [92].

#### 4.3.6. Bladder Cancer

A randomized controlled clinical trial in patients with bladder cancer, NCT04493489, considered the tenth most common cancer in the world and investigated the safety, efficacy, metastasis inhibition capacity, and efficiency in improving patient survival after bladder cancer surgery of propranolol in adjuvant therapy with BCG (Bacillus Calmette–Guerin) [93] (Appendix A). This is a parallel assignment phase 2 with a placebo or control group (with no propranolol co-treatment and following the same treatment protocol as the intervention group). The trial group takes BCG and propranolol orally for 2 consecutive years after transurethral resection of the bladder tumor. The primary endpoint is a recurrence-free survival at two years. No results have been reported so far.

#### 4.3.7. Diverse Tumors

The recognition of new avenues for propranolol in different types of cancer as combined therapy with the standard use of chemotherapeutic agents, COX-2 inhibitors, or other types of drugs is promising and has sparked interest in this compound. The results of ongoing clinical trials and new trials are needed to demonstrate the antitumor benefits of propranolol in combination therapy (Table 1) (Figure 4).

## 5. Conclusions

From the review of registered clinical trials and their resulting literature, several conclusions can be listed regarding the potential use of propranolol as a valid and useful molecule in the treatment of benign and malignant cancers, as follows:-The ADBR1-2 antagonist propranolol has emerged as a candidate treatment for several tumor processes. Although its mechanism of action has yet to be investigated, it is proposed as an adjuvant drug at single administration or in combined therapies due to its safety profile and therapeutic experience that support its use.-The potential therapeutic value of propranolol is supported by its intervention in physiological and molecular mechanisms, such as vasodilatation, apoptosis, or angiogenesis.-Propranolol was tested in several clinical trials for IH at different doses, the timing of treatment, and comparing it to current pharmacological treatments of IH prior to propranolol. In almost all the cases, propranolol showed therapeutic benefits and no side effects.-The EMA gave the marketing authorization to Hemangiol (oral propranolol) after a multicenter phase 3 clinical trial for the treatment of IH. Afterward, propranolol has also been used in monotherapy and combination therapy in different clinical trials to treat different types of tumors.-Among the solid tumors other than IH, the use of propranolol to prevent tumor dissemination and reduce metastasis as a preoperative treatment prior to surgeries in breast, prostate, and ovarian cancer, stands out.

It is important to underline that the majority of the clinical trials with propranolol included a relatively small sample size or are still ongoing (i.e., the results are not yet available). Therefore, future, larger randomized controlled clinical studies specifically focused on propranolol remain necessary.

Despite the limitations of the ongoing clinical trials, the sum of the evidence accumulated for over half a century strongly points to the fact that propranolol could be an ideal team player in combinatorial therapies for different tumor processes. More specifically, it could be used in combination with drugs that interfere with cell division, inflammation, immunotherapy, and antibodies against tumor targets.

## Figures and Tables

**Figure 1 jcm-11-04539-f001:**
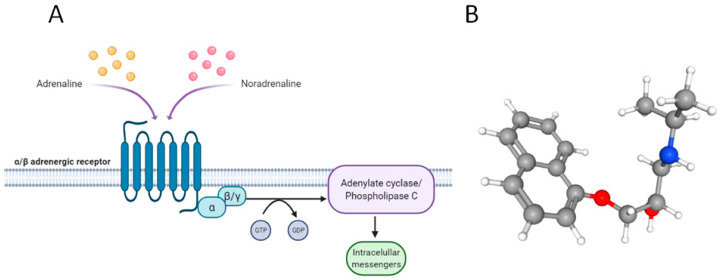
Adrenergic receptors (AR) are classified into two types: α (α1 and α2) and β (β1, β2, and β3), and their ligands are the catecholamines adrenaline and noradrenaline. ARs are coupled to the G-protein whose activation stimulates phospholipase C or adenylate cyclase, promoting the activation of certain genes (**A**). The atomic structure of the propranolol molecule (**B**).

**Figure 2 jcm-11-04539-f002:**
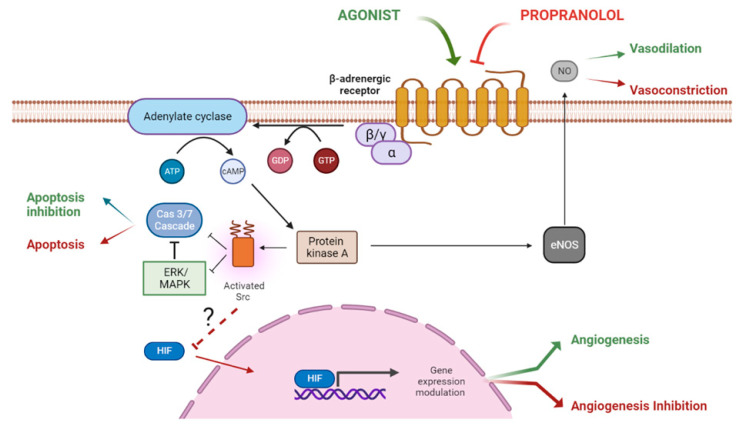
The ADBR signaling in the presence of the ligand, which will be blocked by propranolol. According to this model, propranolol would decrease the adenylate cyclase activity, decreasing the cAMP levels and the activation of PKA. Activation of eNOS by PKA will be reduced, leading to vasoconstriction (1). On the other hand, the decrease in PKA activity will affect src impairing the HIF-1 nuclear translocation with the downregulation of its nuclear targets, among them pro-angiogenic genes, such as VEGF (2). The decreased phosphorylation of the ERK/MAPK kinases cascade and Src will activate the caspase cascade, leading to apoptosis (3).

**Figure 3 jcm-11-04539-f003:**
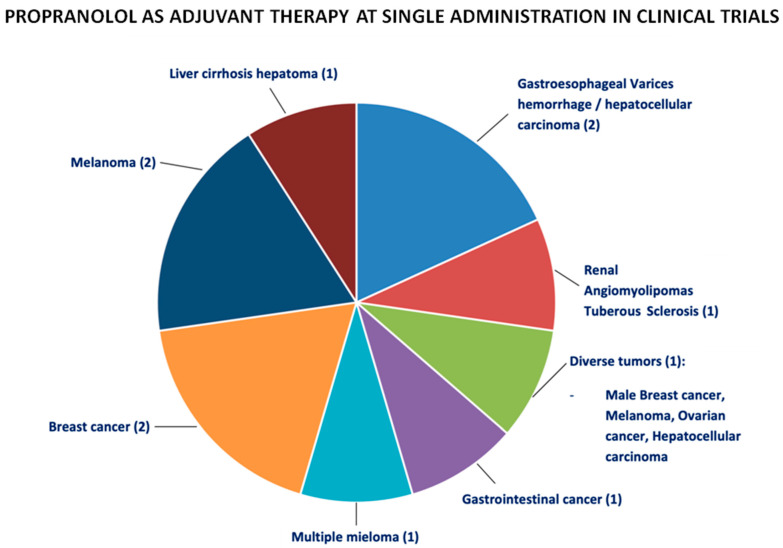
The distribution of the different clinical trials where propranolol has been used as adjuvant therapy at single administration for different types of oncological processes commented on in the main text. In brackets, the number of trials performed.

**Figure 4 jcm-11-04539-f004:**
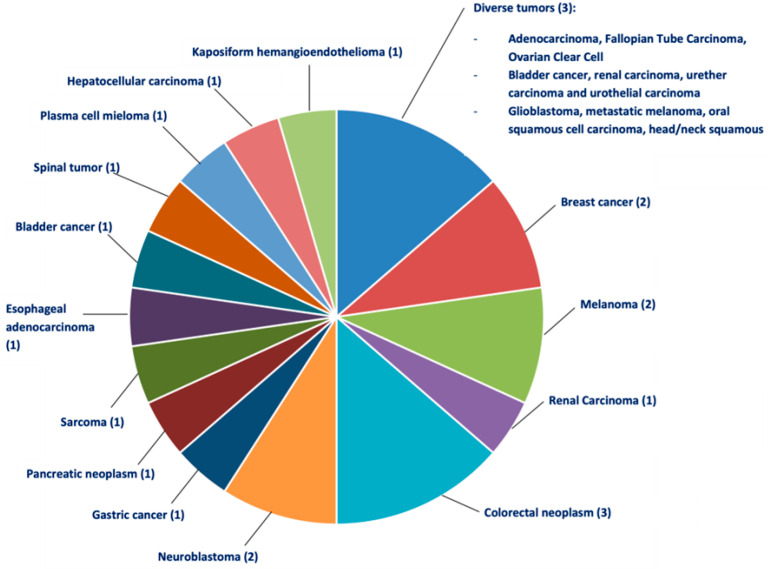
The distribution of the different clinical trials where propranolol has been used in combinatorial treatment therapies. In this case propranolol has been combined with different chemotherapeutic compounds in a wide range of diverse oncological processes. In brackets, the number of tests performed.

**Table 1 jcm-11-04539-t001:** Propranolol as adjuvant to different chemotherapeutic drugs used in combination in interventional clinical trials registered at the EU Clinical Trials Register (https://www.clinicaltrialsregister.eu), the U.S. National Library of Medicine (https://clinicaltrials.gov), and the Australian New Zealand Clinical Trials Registry (http://www.anzctr.org.au/Default.aspx). * Studies with no results published, for further details about the trial, please, visit the Appendix A. Accessed on 14 June 2022.

Drug in Combination	Therapeutic Effect	Type of Cancer	Ref
Bacilli Calmette-Guerin	Immune System activator	Bladder	[93]
Captopril	Angiotensin-converting enzyme (ACE) inhibitor	Infantile Hemangioma	*
Carboplatin	DNA duplication interferent	Esophageal Adenocarcinoma Fallopian Tube Invasive Epithelial Ovarian Primary Peritoneal	[89,90]
Celecoxib cyclophosphamide	COX-2 inhibitor and nonsteroidal anti-inflammatory drug (NSAID)	Neuroblastoma	[86]
Cilazapril	Angiotensin-converting enzyme (ACE) inhibitor	Glioblastoma Head and Neck skin Squamous Cell Metastatic Melanoma Oral cavity Squamous Cell	*
Cyclophosphamide	Alkylating agent	Breast	[91]
Doxorubicin	DNA duplication interferent	Breast	[91]
Etodolac	Nonsteroidal anti-inflammatory drug (NSAID)	Breast Colorectal Pancreatic Neuroblastoma	[79,80,81,82,83,84,85]
Etoposide	Topoisomerase II inhibitor	Neuroblastoma	*
Losartan	Angiotensin II receptor antagonist	Glioblastoma Head and Neck skin Squamous cell carcinoma Metastatic Melanoma Oral cavity Squamous Cell	*
Metformin	Inhibitor of the mitochondrial respiratory chain (complex I)	Glioblastoma Head and Neck skin Squamous Cell Metastatic Melanoma Oral cavity Squamous Cell	*
Paclitaxel	Mitotic inhibitor	Breast Esophageal Adenocarcinoma Fallopian Tube Invasive Epithelial Ovarian Primary Peritoneal	[89,90,91]
Pegfilgrastim	Stimulate the production of neutrophils	Breast	[91]
Pembrolizumab	Binds to and blocks PD-1	Cutaneous Melanoma	[92]
Pertuzumab	HER2 dimerization inhibitor	Breast	[91]
Piperine	Alkaloid	Glioblastoma Head and Neck skin Squamous Cell Metastatic Melanoma Oral cavity Squamous Cell	*
Prednisolone	Immunosuppressor	Kaposiform Hemangioendothelioma Kasabach Merritt Phenomenon	*
Sirolimus	Immunosuppressor	Kaposiform Hemangioendothelioma Kasabach Merritt Phenomenon	*
Trastuzumab	HER2 antagonist	Breast	[91]
Vinblastine	Microtubules assembly inhibitor	Neuroblastoma	*

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
