# Peer review of "Propranolol: A “Pick and Roll” Team Player in Benign Tumors and Cancer Therapies"

_jcm, 2022, doi:10.3390/jcm11154539_

Round 1
Reviewer 1 Report
Review comments
The review article by Virginia Albiñana et al on “Propranolol: A “Pick and Roll” Team Player in Benign Tumors and Cancer Therapies” is an interesting subject. Some of my concerns are:
1. Please write a short description about the history of propranolol. James W. Black won a noble prize in 1988 for discovery of propranolol. The authors should mention everything in detail as well as whether this drug is chemically synthesized or occurring naturally.
2. I feel it is a very good review paper of propranolol and after some minor addition and modification
Author Response
Rebuttal letter: manuscript # jcm-1799453.
Propranolol: A “Pick and Roll” Team Player in Benign Tumors and Cancer Therapies, as a contribution to the Special Issue: Present and Future of Targeted Molecular Therapies in Solid Tumors – Clinician’s Perspective.
Dear Editor and Reviewers,
We thank the editor and the four reviewers for the time they spent looking over the manuscript. We also appreciate the comments and suggestions done to improve quality and the clarity of the manuscript.
Below is our response to each point raised by the reviewers.
All major changes made to the original manuscript have been highlighted in red. Furthermore, the English of the text has been edited by a native speaker to improve the quality of it.
We have also modified tables following the reviewers’ comments.
We ask for the Editor in charge decision about an opposite point of view between one of the reviewers and the authors.
Finally, we hope that we satisfyingly addressed them all and that the manuscript will be now suited for publication.
Sincerely,
On behalf of all authors,
Luisa María Botella and Angel M Cuesta.
Reviewer 1:
The review article by Virginia Albiñana et al on “Propranolol: A “Pick and Roll” Team Player in Benign Tumors and Cancer Therapies” is an interesting subject. Some of my concerns are:
- Please write a short description about the history of propranolol. James W. Black won a noble prize in 1988 for discovery of propranolol. The authors should mention everything in detail as well as whether this drug is chemically synthesized or occurring naturally.
Why propranolol: a short story about propranolol discovery
We appreciate the suggestion, in fact, during the design of this review, we have also been discussing whether it is adequate to include “the origin of propranolol” in a Special Issue named: "Present and Future of Targeted Molecular Therapies in Solid Tumors – Clinician’s Perspective". Nevertheless, we agree that Sir James Whyte Black deserves much more than some paragraphs in a review.
In this sense, we have included (and highlighted in red), in section 2, new paragraphs describing a Historical perspective of propranolol. We hope it fits with what the reviewer had in mind. The text is as follows:
“More than a century ago scientist speculated that catecholamines, after binding selectively to receptor-like structures, caused their pharmacological actions. Was in 1948 when R. P. Ahlquist described, for the first time, the α-and β-adrenotropic receptors (PMID: 18882199). Some years later, Sir James Whyte Black, a Scottish pharmacologist at the Imperial Chemical Industries (ICI) in UK, was searching for a chemically synthesized agent to interfere catecholamines and, therefore, decrease the oxygen heart requirements and relieve the pain of angina. He was searching for a β-adrenergic blocker.
There were two main candidates: pronethalol and propranolol (also named ICI 45,520 or, since 1964, marketed as Inderal). The latter is a more potent derivative from the former and lacks the carcinogenicity founded in animal models treated with pronethalol. Then, propranolol became a best-selling drug, used to treat a wide range of cardiovascular diseases such as arrhythmia, hypertension, and hypertrophic cardiomyopathy. Finally, in 1988, Sir James W. Black was awarded the Nobel Prize in Medicine for creation of propranolol (PMID: 9456487).
Nowadays, propranolol has fallen from the 1st line treatment since it does not perform as well as other drugs. Nevertheless, evidence began to accumulate paving the potential use (reposition) of propranolol as an antitumor drug. This use is supported by several case reports [9–12] that postulated the use of propranolol in a rare tumor like pheochromocytoma to reduce the hypertension generated by catecholamines secretion. Interestingly, these studies outlined (but did not relate) the absence of metastases and tumor growth amelioration without noticeable side effects, together with normalization of heart rate and blood pressure. Few years later, different reports started to show clinical benefits of propranolol in different hyperproliferative diseases such as astrocytomas (PMID: 4330940), epidermoid carcinoma (PMID: 165357 and PMID: 6143691), malignant insulinoma (PMID: 6302161), lung adenocarcinoma (PMID: 2569945), and breast cancer (PMID: 10845278).
Nevertheless, propranolol continued on the bench as cardioprotective medicine until…”
- I feel it is a very good review paper of propranolol and after some minor addition and modification
We are glad that the reviewer liked our work.
We appreciate all the valuable comments from reviewers that have led us to rewrite and improve the manuscript in many aspects.
REFERENCES:
PMID: 165357. Kelly LA, Butcher RW. Studies on cyclic AMP metabolism in human epidermoid carcinoma (HEp-2) cells. Metabolism. 1975 Mar;24(3):359-68. doi: 10.1016/0026-0495(75)90116-x.
PMID: 10845278. Slotkin TA, Zhang J, Dancel R, Garcia SJ, Willis C, Seidler FJ. Beta-adrenoceptor signaling and its control of cell replication in MDA-MB-231 human breast cancer cells. Breast Cancer Res Treat. 2000 Mar;60(2):153-66. doi: 10.1023/a:1006338232150.
PMID: 18882199. AHLQUIST RP. A study of the adrenotropic receptors. Am J Physiol. 1948 Jun;153(3):586-600. doi: 10.1152/ajplegacy.1948.153.3.586.
PMID: 2569945: Schuller, H.M.; Cole, B. Regulation of cell proliferation by β-adrenergjc receptors in a human lung adenocarcinoma cell line. Carcinogenesis 1989, 10, 1753-1755.
PMID: 6143691. Delavier-Klutchko C, Hoebeke J, Strosberg AD. The human carcinoma cell line A431 possesses large numbers of functional beta-adrenergic receptors. FEBS Lett. 1984 Apr 24;169(2):151-5. doi: 10.1016/0014-5793(84)80308-7.
PMID: 6302161. Blum I, Rusecki Y, Doron M, Lahav M, Laron Z, Atsmon A. Evidence for a therapeutic effect of dl-propranolol in benign and malignant insulinoma: report of three cases. J Endocrinol Invest. 1983 Feb;6(1):41-5. doi: 10.1007/BF03350559.
PMID: 9456487. Stapleton MP. Sir James Black and propranolol. The role of the basic sciences in the history of cardiovascular pharmacology. Tex Heart Inst J. 1997;24(4):336-42.

Reviewer 2 Report
Dear authors,
Your review gives interesting information on the role of propranolol in the treatment of cancer. I have some comments:
1. Chapter 2: Please provide more details about the potential mechanism underling anti-angiogenic effects of propranolol
2. Chapter 3.2.1 please also report the results of a recent published systematic review and meta-analysis about the use of betablockers (included propranolol) for patients with triple negative breast cancer which support your conclusions (PMID: 31607128). The authors also provide information about the evidence of propranolol coming from preclinical studies.
3. Given the fact the propranolol is still not yet approved for malignant tumors, why did you decide to do not report any observational studies that might provide evidence of combining propranolol to other chemotherapies? I suggest author to briefly look at published literature an implement the manuscript
4. Line 348: What do you mean with diverse chemotherapeutics? Standard chemotherapy?
5. Table 1. I suggest also to report by each type of cancer the respective references, i.e., melanoma (ref1, ref2). Moreover, please report also the stage of the disease in which it is tested and how it is used (neoadjuvant/adjuvant/ first-line treatment…)
6. I suggest reporting the posology of propranolol in which it is tested for each clinical trial reported in the text and comment more on this aspect in the discussion/conclusion section
Author Response
Rebuttal letter: manuscript # jcm-1799453.
Propranolol: A “Pick and Roll” Team Player in Benign Tumors and Cancer Therapies, as a contribution to the Special Issue: Present and Future of Targeted Molecular Therapies in Solid Tumors – Clinician’s Perspective.
Dear Editor and Reviewers,
We thank the editor and the four reviewers for the time they spent looking over the manuscript. We also appreciate the comments and suggestions done to improve quality and the clarity of the manuscript.
Below is our response to each point raised by the reviewers.
All major changes made to the original manuscript have been highlighted in red. Furthermore, the English of the text has been edited by a native speaker to improve the quality of it.
We have also modified tables following the reviewers’ comments.
We ask for the Editor in charge decision about an opposite point of view between one of the reviewers and the authors.
Finally, we hope that we satisfyingly addressed them all and that the manuscript will be now suited for publication.
Sincerely,
On behalf of all authors,
Luisa María Botella and Angel M Cuesta.
Reviewer 2:
Your review gives interesting information on the role of propranolol in the treatment of cancer. I have some comments:
- Chapter 2: Please provide more details about the potential mechanism underling anti-angiogenic effects of propranolol
We thank the reviewer for the indication. We have added the following paragraphs in section 2.3. “Inhibition of Angiogenesis” (and highlighted in red). “To the best of our knowledge, the complete mechanism of action of propranolol has not yet been revealed, but some steps and facts related with its antiangiogenic properties are beginning to be elucidated. The first description of the consequences (apoptosis) of treating endothelial cells with propranolol was in 2002 (PMID 12418927 and PMID: 12418927), but true antiangiogenic data began in 2009 by Annabi et al., (PMID: 19467330). Since then, in vitro assays with endothelial cells from different tissues such as umbilical cord (human umbilical vein endothelial cells, HUVEC) (PMID: 20732454), brain (human brain microvascular endothelial cells, HBMEC), embryonic stem cells (PMID: 25130141), hemangioma (PMID: 22580939 and PMID: 25728347) and its stem cell derivative, (PMID: 24427325; PMID: 26574555 and PMID: 23096601) and hemangioblastoma from von Hippel-Lindau (VHL) disease (PMID: 26394686 and PMID: 31296894), from neuroblastoma (PMID: 23695022) and white blood cells (leukemic and monocytes) (PMID: 19946348), and in vivo assays on retinopathies (PMID: 20739470.) and vascular lesions (PMID: 22743651) have demonstrated the antiangiogenic effects of propranolol. The research developed on these cells demonstrate that propranolol exerts direct antiangiogenic effects by decreasing the expression levels of key molecules on the angiogenic process such as VGEF (PMID: 26394686; PMID: 19946348; PMID: 20739470; PMID: 22580939; PMID: 24427325; PMID: 25728347 and PMID: 25130141) and its receptor VEGFR (PMID: 20732454; PMID: 22580939; PMID: 25728347 and PMID: 25130141), HIF1-α (PMID: 26394686; PMID: 20739470 and PMID: 25130141), bFGF (PMID: 24427325), FGF-2 (PMID: 25130141), MMP-9 (PMID: 19467330 and PMID: 25728347), MMP-2 (PMID: 20732454 and PMID: 19946348), EPO (PMID: 26394686), p-cofilin (PMID: 25728347), vascular markers as CD31 and VE-cadherin (PMID: 25130141) and vascular regulators as eNOS and NO (PMID: 25130141). Propranolol also interferes in signaling pathways which end in the impairment of the angiogenic process, some of them are the following: cell cycle arrest by downregulation of cyclins A2, D1-3 and upregulation of p15, p21 and p27 (PMID: 20732454 and PMID: 25728347), inhibition of the pI3K/Akt (PMID: 22580939 and PMID: 25728347), p38/MAP (PMID: 22580939) and ERK1/2 signaling pathways (PMID: 20732454).
- Chapter 3.2.1 please also report the results of a recent published systematic review and meta-analysis about the use of betablockers (included propranolol) for patients with triple negative breast cancer which support your conclusions (PMID: 31607128). The authors also provide information about the evidence of propranolol coming from preclinical studies.
We thank the reviewer for the suggestion of the article. During the design of this review, we were discussing whether it is adequate to add in vitro or preclinical data in a Special Issue named: "Present and Future of Targeted Molecular Therapies in Solid Tumors – Clinician’s Perspective". The focus of our manuscript was to only consider propranolol in clinical trials. We are aware that many preclinical studies have been conducted with propranolol, but these studies are out of the scope of this Special Issue.
Nevertheless, as the reviewer certainly points out, the results shown by Spini A et al. (PMID: 31607128), support our conclusions and are in line with the results presented by Montoya A et al. (PMID: 28031536). Unfortunately, and continuing with basketball jargon, Cardwell CR et al., published in 2016 a “ball fake" pooled analysis (PMID: 27906047). Cardwell showed opposite conclusions for propranolol in breast cancer. Nevertheless, this study has not analyzed the data based on the type of breast cancer while Spini’s and Montoya’s analysis had. This difference may result in a bias in the conclusion.
Hence, we think that even that controversial data are in favor of a rational scientific dialogue, preclinical data do not match with the aim of this review and would dramatically extend the length of the manuscript. We consider that such interesting point could be the aim of another revision in depth.
We understand that this is not the expected answer, and we apologize if the reviewer does not agree our point of view. However, we are sincerely opened to include basic research and preclinical studies if both the reviewer and the Editorial Board consider the reviewer’s suggestion.
- Given the fact the propranolol is still not yet approved for malignant tumors, why did you decide to do not report any observational studies that might provide evidence of combining propranolol to other chemotherapies? I suggest author to briefly look at published literature an implement the manuscript
We thank the reviewer for highlighting such a point. We have revisited the clinical trials registers and found 5 observational studies. All but one focus on Infantile Hemangioma, the other one focuses on Pediatric Cancer. Since none of them, even that finished, hadn’t presented results, we have created a new Supplementary Table 3 showing the most relevant data of these observational data.
We have also added in the section “3. Propranolol as Single Treatment in Clinical Trials”, a new subsection “3.4. Propranolol in observational studies”. With the following text (highlighted in red):
“Since propranolol has not yet been approved for malignant tumors, observational studies where propranolol has been used are included in a new Supplementary Table 2. Among them, 4 were related to IH and one for pediatric cancers. NCT04105517 is a phase 4 Hemangiol postmarketing study to look for adverse events in 500 children treated with propranolol for IH. Retrospective NCT04651049, collects parameters of physiological development in children treated with propranolol due to IH. The only observational study including pediatric cancer is NCT02165683, where propranolol was used as an adjuvant with standard chemotherapies for different pediatric cancers. The aim of this study was to reduce the fluorodeoxyglucose (FDG)-positron emission tomography (PET) uptake as an indicator of improvement in the cancer treatment. As seen in the new Supplementary Table 2, no results are available, although the studies have been completed.”
- Line 348: What do you mean with diverse chemotherapeutics? Standard chemotherapy?
We thank the reviewer for the suggestion to clarify this section. We meant Standard chemotherapies based on the type of cancer. Hence, we have changed the title for: “Propranolol and Standard Chemotherapy” (highlighted in red).
- Table 1. I suggest also to report by each type of cancer the respective references, i.e., melanoma (ref1, ref2). Moreover, please report also the stage of the disease in which it is tested and how it is used (neoadjuvant/adjuvant/ first-line treatment…)
This information is provided in Supplementary Table 3 (which was previously named Supplementary Table 2). We have clarified this fact in Table 1, which is a summary of the different adjuvants and treatments used with propranolol in clinical trials. The complete information of these trials can be found in the Supplementary Table 3, the reason is not to extend in excess the length of the manuscript. However, if the reviewer considers its inclusion as main tables in the text, we will accept it.
- I suggest reporting the posology of propranolol in which it is tested for each clinical trial reported in the text and comment more on this aspect in the discussion/conclusion section (dosage of propranolol, and discuss at the end the dosage)
Thanks for the suggestion. We have added a new column in the tables (and highlighted in red) indicating the dosage of every trial, when provided.
Finally, we appreciate all the valuable comments from reviewers that have led us to rewrite and improve the manuscript in many aspects.
REFERENCES:
PMID: 10845278. Slotkin TA, Zhang J, Dancel R, Garcia SJ, Willis C, Seidler FJ. Beta-adrenoceptor signaling and its control of cell replication in MDA-MB-231 human breast cancer cells. Breast Cancer Res Treat. 2000 Mar;60(2):153-66. doi: 10.1023/a:1006338232150.
PMID: 12418927. Sommers Smith SK, Smith DM. Beta blockade induces apoptosis in cultured capillary endothelial cells. In Vitro Cell Dev Biol Anim. 2002 May;38(5):298-304. doi: 10.1290/1071-2690(2002)038<0298:BBIAIC>2.0.CO;2.
PMID: 165357. Kelly LA, Butcher RW. Studies on cyclic AMP metabolism in human epidermoid carcinoma (HEp-2) cells. Metabolism. 1975 Mar;24(3):359-68. doi: 10.1016/0026-0495(75)90116-x.
PMID: 18882199. AHLQUIST RP. A study of the adrenotropic receptors. Am J Physiol. 1948 Jun;153(3):586-600. doi: 10.1152/ajplegacy.1948.153.3.586.
PMID: 19467330. Annabi B, Lachambre MP, Plouffe K, Moumdjian R, Béliveau R. Propranolol adrenergic blockade inhibits human brain endothelial cells tubulogenesis and matrix metalloproteinase-9 secretion. Pharmacol Res. 2009 Nov;60(5):438-45. doi: 10.1016/j.phrs.2009.05.005. Epub 2009 May 23.
PMID: 19946348. Hajighasemi F, Hajighasemi S. Effect of propranolol on angiogenic factors in human hematopoietic cell lines in vitro. Iran Biomed J. 2009 Oct;13(4):223-8.
PMID: 20732454. Lamy S, Lachambre MP, Lord-Dufour S, Béliveau R. Propranolol suppresses angiogenesis in vitro: inhibition of proliferation, migration, and differentiation of endothelial cells. Vascul Pharmacol. 2010 Nov-Dec;53(5-6):200-8. doi: 10.1016/j.vph.2010.08.002. Epub 2010 Aug 20.
PMID: 20739470. Ristori C, Filippi L, Dal Monte M, Martini D, Cammalleri M, Fortunato P, la Marca G, Fiorini P, Bagnoli P. Role of the adrenergic system in a mouse model of oxygen-induced retinopathy: antiangiogenic effects of beta-adrenoreceptor blockade. Invest Ophthalmol Vis Sci. 2011 Jan 5;52(1):155-70. doi: 10.1167/iovs.10-5536.
PMID: 22580939. Chim H, Armijo BS, Miller E, Gliniak C, Serret MA, Gosain AK. Propranolol induces regression of hemangioma cells through HIF-1α-mediated inhibition of VEGF-A. Ann Surg. 2012 Jul;256(1):146-56. doi: 10.1097/SLA.0b013e318254ce7a.
PMID: 22743651. Chisholm KM, Chang KW, Truong MT, Kwok S, West RB, Heerema-McKenney AE. β-Adrenergic receptor expression in vascular tumors. Mod Pathol. 2012 Nov;25(11):1446-51. doi: 10.1038/modpathol.2012.108. Epub 2012 Jun 29.
PMID: 23096601. Wong A, Hardy KL, Kitajewski AM, Shawber CJ, Kitajewski JK, Wu JK. Propranolol accelerates adipogenesis in hemangioma stem cells and causes apoptosis of hemangioma endothelial cells. Plast Reconstr Surg. 2012 Nov;130(5):1012-1021. doi: 10.1097/PRS.0b013e318267d3db.
PMID: 23695022. Pasquier E, Street J, Pouchy C, Carre M, Gifford AJ, Murray J, Norris MD, Trahair T, Andre N, Kavallaris M. β-blockers increase response to chemotherapy via direct antitumour and anti-angiogenic mechanisms in neuroblastoma. Br J Cancer. 2013 Jun 25;108(12):2485-94. doi: 10.1038/bjc.2013.205. Epub 2013 May 21.
PMID: 24427325. Zhang L, Mai HM, Zheng J, Zheng JW, Wang YA, Qin ZP, Li KL. Propranolol inhibits angiogenesis via down-regulating the expression of vascular endothelial growth factor in hemangioma derived stem cell. Int J Clin Exp Pathol. 2013 Dec 15;7(1):48-55.
PMID: 25130141. Sharifpanah F, Saliu F, Bekhite MM, Wartenberg M, Sauer H. β-Adrenergic receptor antagonists inhibit vasculogenesis of embryonic stem cells by downregulation of nitric oxide generation and interference with VEGF signalling. Cell Tissue Res. 2014 Nov;358(2):443-52. doi: 10.1007/s00441-014-1976-8. Epub 2014 Aug 19.
PMID: 2569945: Schuller, H.M.; Cole, B. Regulation of cell proliferation by β-adrenergjc receptors in a human lung adenocarcinoma cell line. Carcinogenesis 1989, 10, 1753-1755.
PMID: 25728347. Pan WK, Li P, Guo ZT, Huang Q, Gao Y. Propranolol induces regression of hemangioma cells via the down-regulation of the PI3K/Akt/eNOS/VEGF pathway. Pediatr Blood Cancer. 2015 Aug;62(8):1414-20. doi: 10.1002/pbc.25453. Epub 2015 Mar 1.
PMID: 26394686. Albiñana, V.; Villar Gómez De Las Heras, K.; Serrano-Heras, G.; Segura, T.; Perona-Moratalla, A.B.; Mota-Pérez, M.; de Campos, J.M.; Botella, L.M. Propranolol Reduces Viability and Induces Apoptosis in Hemangioblastoma Cells from von Hippel-Lindau Patients. Orphanet J Rare Dis 2015, 10, 118.
PMID: 26574555. Munabi, N.C.; England, R.W., Edwards, A.K., Kitajewski, A.A.; Tan, Q.K.; Weinstein, A.; Kung, J.E.; Wilcox, M.; Kitajewski, J.K.; Shawber, C.J.; Wu, J.K. Propranolol Targets Hemangioma Stem Cells via cAMP and Mitogen-Activated Protein Kinase Regulation. Stem Cells Transl Med 2016, 5, 45-55.
PMID: 27085011. Khouri C, Jouve T, Blaise S, Carpentier P, Cracowski JL, Roustit M. Peripheral vasoconstriction induced by β-adrenoceptor blockers: a systematic review and a network meta-analysis. Br J Clin Pharmacol. 2016 Aug;82(2):549-60. doi: 10.1111/bcp.12980. Epub 2016 May 31.
PMID: 27906047. Cardwell CR, Pottegård A, Vaes E, Garmo H, Murray LJ, Brown C, Vissers PA, O'Rorke M, Visvanathan K, Cronin-Fenton D, De Schutter H, Lambe M, Powe DG, van Herk-Sukel MP, Gavin A, Friis S, Sharp L, Bennett K. Propranolol and survival from breast cancer: a pooled analysis of European breast cancer cohorts. Breast Cancer Res. 2016 Dec 1;18(1):119. doi: 10.1186/s13058-016-0782-5.
PMID: 28031536. Montoya A, Amaya CN, Belmont A, Diab N, Trevino R, Villanueva G, Rains S, Sanchez LA, Badri N, Otoukesh S, Khammanivong A, Liss D, Baca ST, Aguilera RJ, Dickerson EB, Torabi A, Dwivedi AK, Abbas A, Chambers K, Bryan BA, Nahleh Z. Use of non-selective β-blockers is associated with decreased tumor proliferative indices in early stage breast cancer. Oncotarget. 2017 Jan 24;8(4):6446-6460. doi: 10.18632/oncotarget.14119.
PMID: 31296894. Cuesta, A.M.; Albiñana, V.; Gallardo-Vara, E.; Recio-Poveda, L.; de Rojas-P, I.; de Las Heras, K.V.G.; Aguirre, D.T.; Botella, L.M. The Β2-Adrenergic Receptor Antagonist ICI-118,551 Blocks the Constitutively Activated HIF Signalling in Hemangioblastomas from von Hippel-Lindau Disease. Sci Rep 2019, 9, 10062.
PMID: 31607128: Spini, A.; Roberto, G.; Gini, R.; Bartolini, C.; Bazzani, L.; Donnini, S.; Crispino, S.; Ziche, M. Evidence of β-blockers drug repurposing for the treatment of triple negative breast cancer: A systematic review. Neoplasma 2019, 66, 963-970.
PMID: 35784377. Wilson S, Hassan D, Jakeman M, Breuning E. Cost-effectiveness of atenolol compared to propranolol as first-line treatment of infantile haemangioma: A pilot study. JPRAS Open. 2022 May 26;33:52-56. doi: 10.1016/j.jpra.2022.05.010.
PMID: 4330940. Clark RB, Perkins JP. Regulation of adenosine 3':5'-cyclic monophosphate concentration in cultured human astrocytoma cells by catecholamines and histamine. Proc Natl Acad Sci U S A. 1971 Nov;68(11):2757-60. doi: 10.1073/pnas.68.11.2757.
PMID: 6143691. Delavier-Klutchko C, Hoebeke J, Strosberg AD. The human carcinoma cell line A431 possesses large numbers of functional beta-adrenergic receptors. FEBS Lett. 1984 Apr 24;169(2):151-5. doi: 10.1016/0014-5793(84)80308-7.
PMID: 6302161. Blum I, Rusecki Y, Doron M, Lahav M, Laron Z, Atsmon A. Evidence for a therapeutic effect of dl-propranolol in benign and malignant insulinoma: report of three cases. J Endocrinol Invest. 1983 Feb;6(1):41-5. doi: 10.1007/BF03350559.
PMID: 9456487. Stapleton MP. Sir James Black and propranolol. The role of the basic sciences in the history of cardiovascular pharmacology. Tex Heart Inst J. 1997;24(4):336-42.

Reviewer 3 Report
The review highlights potential anti-tumor effects of propranolol, through several mechanisms. However, describing it as “pick and roll” in treatment strategies is not justified.
Figure 1A does not include any data about the potential signaling pathways related to carcinogenesis as referred to in the text.
No need for the abbreviation MVP “Most valuable player”, as it was not repeated in the manuscript.
Lines 82 and 83: Propranolol is used in pheochromocytoma together with alpha blockers to treat hypertension and NOT as an anti-tumor agent. The statement and assumption is incorrect.
What is MPOZ?
The manuscript includes irrationalities due to presenting only part of the mechanism of action of propranolol, e.g., considering propranolol as a vasoconstrictor is not correct. The drug mediates central actions that leads to lowering of blood pressure, including blockade beta1 receptors in the kidneys, with resultant suppression of the renin-angiotensin aldosterone system (RAAS), that definitely lead to vasodilation and mediates the sustained antihypertensive effect of propranolol (and other beta blockers). The beta 2 effect on the blood vessels is overcome by the suppression of RAAS, thus the claimed vasoconstrictor effect of propranolol is initial and transient, if any. It could not be considered as a robust mechanism that may decrease the tumor vascularity as stated in the manuscript.
Line #185: Atenolol is a selective beta 1 blocker, with no vasoconstrictive effect. Here, the mechanism of action and the rationale of its use to treat IH should be explained. Having captopril mentioned in table 1 in treatment of IH, it seems that the hypotensive effects of both ACE inhibitors and beta blockers lead to ameliorating of the IH, rather than the assumed vasoconstrictor effect.
The title of section 3: “Propranolol as Single Therapy in Clinical Trials” applies only to subtitle 3.1. Infantile hemangioma, not to further subtitles, where propranolol is used mainly as an adjuvant therapy to increase the tumor sensitivity to anticancer therapies.
Table 1 gives a false impression that anticancer medications (among other medications) are used in combination with propranolol as adjuvant therapies. In fact, propranolol is used as an adjuvant therapy is those trials and its specific mechanism of action in each tumor should be described. The number of the clinical trial and (NCT) and the references if the results are available should be added in a separate column.
Section 3.2.2. and 3.2.3 on “Melanoma” and “other tumors”should be deleted, as no information is available on the clinical trials, as the authors mentioned.
Title of figure 3 should be corrected to “adjuvant” and NOT “single” therapy.
Section 4: Propranolol in Combinatorial Therapy: for each trial, the rationale of using propranolol should be clearly stated, as was done in Multiple Myeloma. Through this addition, the authors will realize that propranolol is used in different context for reasons other than the assumed direct anti-tumor effect.
The review could be re-written in a different perspective, by objectively compiling and explaining the diverse roles of propranolol in medical oncology. Note that the apoptotic and antiangiogenic effect of propranolol is trivial compared to other anti-cancer agents and targeted therapies. The only significant effect is in IH, which is a benign tumor that usually undergoes involution with time.
Author Response
Rebuttal letter: manuscript # jcm-1799453.
Propranolol: A “Pick and Roll” Team Player in Benign Tumors and Cancer Therapies, as a contribution to the Special Issue: Present and Future of Targeted Molecular Therapies in Solid Tumors – Clinician’s Perspective.
Dear Editor and Reviewers,
We thank the editor and the four reviewers for the time they spent looking over the manuscript. We also appreciate the comments and suggestions done to improve quality and the clarity of the manuscript.
Below is our response to each point raised by the reviewers.
All major changes made to the original manuscript have been highlighted in red. Furthermore, the English of the text has been edited by a native speaker to improve the quality of it.
We have also modified tables following the reviewers’ comments.
We ask for the Editor in charge decision about an opposite point of view between one of the reviewers and the authors.
Finally, we hope that we satisfyingly addressed them all and that the manuscript will be now suited for publication.
Sincerely,
On behalf of all authors,
Luisa María Botella and Angel M Cuesta.
Reviewer 3:
The review highlights potential anti-tumor effects of propranolol, through several mechanisms. However, describing it as “pick and roll” in treatment strategies is not justified.
We appreciate the reviewer's comment.
In this review, we want to emphasize that propranolol is a good team player in cancer therapies, not a franchise player. The pun using basketball jargon is to show that from an initial strategy (a block of the defender to leave the teammate free to act), it has evolved into several different combinations with a single purpose: to facilitate the dunk, to kill the tumor cells.
As we explained in the "Introduction" and "Why propranolol?" sections, propranolol is beginning to show more clinical benefits than expected ab initio and this success encourages for further basic and preclinical research and human.
Finally, we would like to keep "pick and roll" in the title, but we are sincerely opened to remove it if both the reviewer and the Editorial Board consider the reviewer’s suggestion.
Figure 1A does not include any data about the potential signaling pathways related to carcinogenesis as referred to in the text.
Thanks for the comment, we have revised the text of Figure 1 in order to not misleading.
Figure 1 only shows the signaling pathway of the ADBRs and the caption explains the types of ADBRs and the pathway until the cAMP as the first messenger.
The data about the potential signaling pathways related to carcinogenesis are shown in Fig 2 and described in the legend as Figure 2 as follows: “The ADBR signaling in the presence of the ligand, which will be blocked by propranolol. According to this model propranolol would decrease de adenylate cyclase activity, decreasing the cAMP levels and the activation of PKA. Activation of eNOS by PKA will be reduced leading to vasoconstriction (1). On the other hand, the decrease in PKA activity will affect src impairing the HIF-1 nuclear translocation with the downregulation of its nuclear targets among them pro-angiogenic genes as VEGF (2). The decreased phosphorylation of ERK/MAPK kinases cascade and Src will activate the caspase cascade leading to apoptosis (3).
We hope this explanation has cleared the reviewer doubts.
No need for the abbreviation MVP “Most valuable player”, as it was not repeated in the manuscript.
Thank you, we have deleted the acronym.
Lines 82 and 83: Propranolol is used in pheochromocytoma together with alpha blockers to treat hypertension and NOT as an anti-tumor agent. The statement and assumption is incorrect.
This part of the manuscript has been deeply rewritten and the statement/assumption is not present as it was phrased before. The changes have been highlighted in red and here we show you the part of the text in which this assumption has been modified: “This use is supported by several case reports [9–12] that postulated the use of propranolol in a rare tumor like pheochromocytoma to reduce the hypertension generated by catecholamines secretion. Interestingly, these studies outlined (but did not relate) the absence of metastases and tumor growth amelioration without noticeable side effects, together with normalization of heart rate and blood pressure”.
What is MPOZ?
We apologize for not including the meaning of this acronym. MPOZ stand for a special malignant and unresectable kind of pheochromocytoma (a rare tumor) and means Malignant Pheochromocytoma of the Organ of Zuckerkandl. We have included (and highlighted in red) the longer name.
Nevertheless, as we said above, this part has been deeply modified and the term MPOZ has been deleted form the text.
The manuscript includes irrationalities due to presenting only part of the mechanism of action of propranolol, e.g., considering propranolol as a vasoconstrictor is not correct. The drug mediates central actions that leads to lowering of blood pressure, including blockade beta1 receptors in the kidneys, with resultant suppression of the renin-angiotensin aldosterone system (RAAS), that definitely lead to vasodilation and mediates the sustained antihypertensive effect of propranolol (and other beta blockers). The beta 2 effect on the blood vessels is overcome by the suppression of RAAS, thus the claimed vasoconstrictor effect of propranolol is initial and transient, if any. It could not be considered as a robust mechanism that may decrease the tumor vascularity as stated in the manuscript.
Thank you very much for your precision and we regret that the reviewer finds irrationalities in the manuscript.
The mechanism mentioned by the reviewer is correct but is not the only one. We mention propranolol as a peripheral vasoconstrictor since propranolol (of ADBR-blockade) interferes eNOS and, hence decreases in the NO production, a well-known vasodilator (PMID: 27085011 and PMID: 25130141). In addition, in figure 2, the pathway leading to eNOS is included, as affected by propranolol blockade.
Line #185: Atenolol is a selective beta 1 blocker, with no vasoconstrictive effect. Here, the mechanism of action and the rationale of its use to treat IH should be explained. Having captopril mentioned in table 1 in treatment of IH, it seems that the hypotensive effects of both ACE inhibitors and beta blockers lead to ameliorating of the IH, rather than the assumed vasoconstrictor effect.
Thanks for pointing out Atenolol as surrogate drug for propranolol.
The mentioned clinical trial (NCT02342275) is the only one in which Atenolol shows clinical benefits for IH infants. This result has not been replicated in any other clinical trials and, on the other hand, the differences may lie in the fact of being infants “with problematic IH”, suggesting that oral Atenolol could be used as an alternative. It must be also noted that a pilot study analyses the ratio cost-effectiveness lies in favor of Atenolol rather than propranolol PMID: 35784377.
Finally, as commented above, we believe that the vasoconstrictive is happening, as also described in (PMID: 27085011 and PMID: 25130141).
The title of section 3: “Propranolol as Single Therapy in Clinical Trials” applies only to subtitle 3.1. Infantile hemangioma, not to further subtitles, where propranolol is used mainly as an adjuvant therapy to increase the tumor sensitivity to anticancer therapies.
Thanks for this comment.
In order to not mislead the meaning of single therapy and even that, in most of the trials, propranolol is the only drug or therapy used, we keep all the trials in which propranolol was administered alone and also, we have modified the title to “Propranolol as Single Treatment in Clinical Trials”.
Table 1 gives a false impression that anticancer medications (among other medications) are used in combination with propranolol as adjuvant therapies. In fact, propranolol is used as an adjuvant therapy is those trials and its specific mechanism of action in each tumor should be described. The number of the clinical trial and (NCT) and the references if the results are available should be added in a separate column.
We thank this comment. Table 1 is a summary and Table S3 (previously named Supplementary Table 2), shows a more detailed description of the trials. We have included the following sentence in the caption of Table 1: “For further details, please, visit the Supplementary Table 3” (highlighted in red).
Section 3.2.2. and 3.2.3 on “Melanoma” and “other tumors” should be deleted, as no information is available on the clinical trials, as the authors mentioned.
Sections 3.2.2 and 3.2.3 have been deleted but we will keep them in the Table S2 since they are an example of the expectations created by propranolol.
Title of figure 3 should be corrected to “adjuvant” and NOT “single” therapy.
We thank the reviewer’s appreciation; the text has been modified to “Propranolol as Single Treatment in Clinical Trials”.
Section 4: Propranolol in Combinatorial Therapy: for each trial, the rationale of using propranolol should be clearly stated, as was done in Multiple Myeloma. Through this addition, the authors will realize that propranolol is used in different context for reasons other than the assumed direct anti-tumor effect.
We thank this reviewer comment. Section 4 has an introductory paragraph explaining the rationale of using propranolol as an adjuvant in combination with the different drug combinations:
“Early studies using propranolol in combination with other reagents, apart from the case reports already discuss in pheochromocytoma, sought to reduce the preoperative stress in rats and thereby enhance the antitumor response of the chemotherapeutic agent (synergistic effect). The hypothesis was that secreted catecholamines and prostaglandins, under stress conditions: i) impair the host cell mediated immunity (CMI) driven by NK cells, ii) trigger inflammatory processes leading to angiogenic responses, and iii) fuel tumor growth [69–72]. Therefore, a preoperational combination of an ADRB-blockade with propranolol plus treatment with the COX-2 inhibitor Celecoxib would reduce lung metastases, enhance the antitumor activity of the NK cells, and increase the survival rates in different xenografts, as has already occurred [69,73,60,74].
Furthermore, new trials have been supporting the synergistic effect of propranolol as an adjuvant in combination therapies with the cytotoxic 5-FU [23], the TKI sunitinib [75], an anti-PD-1 [76] plus IL-2, the electron transport chain inhibitor metformin [77], and the glycolysis inhibitor 2DG [78].
All this has inspired an increasing number of clinical trials investigating the use of propranolol, as an antiangiogenic drug, in combination with some other anticancer drugs used as a plausible tumor therapy in pre -and post-operative patients. Some of these clinical trials described below appear to be promising combination therapies for different angiogenesis-related diseases, although most of them have either not been completed or have not yet shown conclusive results”.
The review could be re-written in a different perspective, by objectively compiling and explaining the diverse roles of propranolol in medical oncology. Note that the apoptotic and antiangiogenic effect of propranolol is trivial compared to other anti-cancer agents and targeted therapies. The only significant effect is in IH, which is a benign tumor that usually undergoes involution with time.
We appreciate the suggestion. We just followed the scope of this Special Issue named: "Present and Future of Targeted Molecular Therapies in Solid Tumors – Clinician’s Perspective". We think that pointing out the already known anticancer therapies will be just “another review” but showing the potential of combinatorial therapies using RTKIs plus ADBR-blockers can provide new clues in tumor resistance, metastases, and stemness.
In these days of the 3Rs (Reducing, Recycling and Reusing), drug reposition fits completely and lower the costs and the time of the preclinical studies. Therefore, all the light we can put on the cancer therapies research is never enough. As said above, propranolol is a good team player in cancer therapies, not a franchise player that deserves an opportunity in cancer research.
Finally, we appreciate all the valuable comments from reviewers that have led us to rewrite and improve the manuscript in many aspects.
REFERENCES
PMID: 10845278. Slotkin TA, Zhang J, Dancel R, Garcia SJ, Willis C, Seidler FJ. Betaadrenoceptor signaling and its control of cell replication in MDA-MB-231 human breast cancer cells. Breast Cancer Res Treat. 2000 Mar;60(2):153-66. doi: 10.1023/a:1006338232150.
PMID: 12418927. Sommers Smith SK, Smith DM. Beta blockade induces apoptosis in cultured capillary endothelial cells. In Vitro Cell Dev Biol Anim. 2002 May;38(5):298-304. doi: 10.1290/1071-2690(2002)038<0298:BBIAIC>2.0.CO;2.
PMID: 165357. Kelly LA, Butcher RW. Studies on cyclic AMP metabolism in human epidermoid carcinoma (HEp-2) cells. Metabolism. 1975 Mar;24(3):359-68. doi: 10.1016/0026-0495(75)90116-x.
PMID: 18882199. AHLQUIST RP. A study of the adrenotropic receptors. Am J Physiol. 1948 Jun;153(3):586-600. doi: 10.1152/ajplegacy.1948.153.3.586.
PMID: 19467330. Annabi B, Lachambre MP, Plouffe K, Moumdjian R, Béliveau R. Propranolol adrenergic blockade inhibits human brain endothelial cells tubulogenesis and matrix metalloproteinase-9 secretion. Pharmacol Res. 2009 Nov;60(5):438-45. doi:10.1016/j.phrs.2009.05.005. Epub 2009 May 23.
PMID: 19946348. Hajighasemi F, Hajighasemi S. Effect of propranolol on angiogenic factors in human hematopoietic cell lines in vitro. Iran Biomed J. 2009 Oct;13(4):223-8.
PMID: 20732454. Lamy S, Lachambre MP, Lord-Dufour S, Béliveau R. Propranolol suppresses angiogenesis in vitro: inhibition of proliferation, migration, and differentiation of endothelial cells. Vascul Pharmacol. 2010 Nov Dec;53(5-6):200-8. doi: 10.1016/j.vph.2010.08.002. Epub 2010 Aug 20.
PMID: 20739470. Ristori C, Filippi L, Dal Monte M, Martini D, Cammalleri M, Fortunato P, la Marca G, Fiorini P, Bagnoli P. Role of the adrenergic system in a mouse model of oxygeninduced retinopathy: antiangiogenic effects of beta-adrenoreceptor blockade. Invest Ophthalmol Vis Sci. 2011 Jan 5;52(1):155-70. doi: 10.1167/iovs.10-5536.
PMID: 22580939. Chim H, Armijo BS, Miller E, Gliniak C, Serret MA, Gosain AK. Propranolol induces regression of hemangioma cells through HIF-1α-mediated inhibition of VEGF-A. Ann Surg. 2012 Jul;256(1):146-56. doi: 10.1097/SLA.0b013e318254ce7a.
PMID: 22743651. Chisholm KM, Chang KW, Truong MT, Kwok S, West RB, Heerema-McKenney AE. β-Adrenergic receptor expression in vascular tumors. Mod Pathol. 2012 Nov;25(11):1446-51. doi: 10.1038/modpathol.2012.108. Epub 2012 Jun 29.
PMID: 23096601. Wong A, Hardy KL, Kitajewski AM, Shawber CJ, Kitajewski JK, Wu JK.Propranolol accelerates adipogenesis in hemangioma stem cells and causes apoptosis ofhemangioma endothelial cells. Plast Reconstr Surg. 2012 Nov;130(5):1012-1021. doi:10.1097/PRS.0b013e318267d3db.
PMID: 23695022. Pasquier E, Street J, Pouchy C, Carre M, Gifford AJ, Murray J, Norris MD,Trahair T, Andre N, Kavallaris M. β-blockers increase response to chemotherapy via directantitumour and anti-angiogenic mechanisms in neuroblastoma. Br J Cancer. 2013 Jun25;108(12):2485-94. doi:10.1038/bjc.2013.205. Epub 2013 May 21.
PMID: 24427325. Zhang L, Mai HM, Zheng J, Zheng JW, Wang YA, Qin ZP, Li KL. Propranolol inhibits angiogenesis via down-regulating the expression of vascular endothelial growth factor in hemangioma derived stem cell. Int J Clin Exp Pathol. 2013 Dec 15;7(1):48-55.
PMID: 25130141. Sharifpanah F, Saliu F, Bekhite MM, Wartenberg M, Sauer H. β-Adrenergic receptor antagonists inhibit vasculogenesis of embryonic stem cells by downregulation of nitric oxide generation and interference with VEGF signalling. Cell Tissue Res. 2014 Nov;358(2):443-52. doi:10.1007/s00441-014-1976-8. Epub 2014 Aug 19.
PMID: 2569945: Schuller, H.M.; Cole, B. Regulation of cell proliferation by β-adrenergjc receptors in a human lung adenocarcinoma cell line. Carcinogenesis 1989, 10, 1753-1755.
PMID: 25728347. Pan WK, Li P, Guo ZT, Huang Q, Gao Y. Propranolol induces regression of hemangioma cells via the down-regulation of the PI3K/Akt/eNOS/VEGF pathway. Pediatr Blood Cancer. 2015 Aug;62(8):1414-20. doi: 10.1002/pbc.25453. Epub 2015 Mar 1.
PMID: 26394686. Albiñana, V.; Villar Gómez De Las Heras, K.; Serrano-Heras, G.; Segura, T.; Perona-Moratalla, A.B.; Mota-Pérez, M.; de Campos, J.M.; Botella, L.M. Propranolol Reduces Viability and Induces Apoptosis in Hemangioblastoma Cells from von Hippel-Lindau Patients. Orphanet J Rare Dis 2015, 10, 118.
PMID: 26574555. Munabi, N.C.; England, R.W., Edwards, A.K., Kitajewski, A.A.; Tan, Q.K.; Weinstein, A.; Kung, J.E.; Wilcox, M.; Kitajewski, J.K.; Shawber, C.J.; Wu, J.K. Propranolol Targets Hemangioma Stem Cells via cAMP and Mitogen-Activated Protein Kinase Regulation. Stem Cells Transl Med 2016, 5, 45-55.
PMID: 27085011. Khouri C, Jouve T, Blaise S, Carpentier P, Cracowski JL, Roustit M. Peripheral vasoconstriction induced by β-adrenoceptor blockers: a systematic review and a network meta-analysis. Br J Clin Pharmacol. 2016 Aug;82(2):549-60. doi: 10.1111/bcp.12980. Epub 2016 May 31.
PMID: 27906047. Cardwell CR, Pottegård A, Vaes E, Garmo H, Murray LJ, Brown C, Vissers PA, O'Rorke M, Visvanathan K, Cronin-Fenton D, De Schutter H, Lambe M, Powe DG, van Herk Sukel MP, Gavin A, Friis S, Sharp L, Bennett K. Propranolol and survival from breast cancer: a pooled analysis of European breast cancer cohorts. Breast Cancer Res. 2016 Dec 1;18(1):119. doi: 10.1186/s13058-016-0782-5.
PMID: 28031536. Montoya A, Amaya CN, Belmont A, Diab N, Trevino R, Villanueva G, Rains S, Sanchez LA, Badri N, Otoukesh S, Khammanivong A, Liss D, Baca ST, Aguilera RJ, Dickerson EB, Torabi A, Dwivedi AK, Abbas A, Chambers K, Bryan BA, Nahleh Z. Use of non-selective β-blockers is associated with decreased tumor proliferative indices in early stage breast cancer. Oncotarget. 2017 Jan 24;8(4):6446-6460. doi: 10.18632/oncotarget.14119.
PMID: 31296894. Cuesta, A.M.; Albiñana, V.; Gallardo-Vara, E.; Recio-Poveda, L.; de Rojas-P, I.; de Las Heras, K.V.G.; Aguirre, D.T.; Botella, L.M. The Β2-Adrenergic Receptor Antagonist ICI-118,551 Blocks the Constitutively Activated HIF Signalling in Hemangioblastomas from von Hippel-Lindau Disease. Sci Rep 2019, 9, 10062.
PMID: 31607128: Spini, A.; Roberto, G.; Gini, R.; Bartolini, C.; Bazzani, L.; Donnini, S.; Crispino, S.; Ziche, M. Evidence of β-blockers drug repurposing for the treatment of triple negative breast cancer: A systematic review. Neoplasma 2019, 66, 963-970.
PMID: 35784377. Wilson S, Hassan D, Jakeman M, Breuning E. Cost-effectiveness of atenolol compared to propranolol as first-line treatment of infantile haemangioma: A pilot study. JPRAS Open. 2022 May 26;33:52-56. doi: 10.1016/j.jpra.2022.05.010.
PMID: 4330940. Clark RB, Perkins JP. Regulation of adenosine 3':5'-cyclic monophosphate concentration in cultured human astrocytoma cells by catecholamines and histamine. Proc Natl Acad Sci U S A. 1971 Nov;68(11):2757-60. doi: 10.1073/pnas.68.11.2757.
PMID: 6143691. Delavier-Klutchko C, Hoebeke J, Strosberg AD. The human carcinoma cell line A431 possesses large numbers of functional beta-adrenergic receptors. FEBS Lett. 1984 Apr 24;169(2):151-5. doi: 10.1016/0014-5793(84)80308-7.
PMID: 6302161. Blum I, Rusecki Y, Doron M, Lahav M, Laron Z, Atsmon A. Evidence for a therapeutic effect of dl-propranolol in benign and malignant insulinoma: report of three cases. J Endocrinol Invest. 1983 Feb;6(1):41-5. doi: 10.1007/BF03350559.
PMID: 9456487. Stapleton MP. Sir James Black and propranolol. The role of the basic sciences in the history of cardiovascular pharmacology. Tex Heart Inst J. 1997;24(4):336-42.

Reviewer 4 Report
Dear Editor,
Within their manuscript, the authors give an interesting overview about actual information about propanolol with a focus on its clinical application in cancer treatment. In addition, they report all the clinical trials on propranolol use. The information in the manuscript can be helpful in recapitulating some aspects, pointing in specific critical points.
The literature research did not show any similar review like this one, so any plagiarism was detected.
The structure and layout of the manuscript were written in an organized way, easy to read and understand; however, sometime, there are no references of in vitro studies evaluating the effect of propanolol on cancer cells. I fully understand that this Journal call for the clinical topic but to improve the quality of the manuscript, I believe should be important insert some works rely on the effectiveness of propanolol on cancer cells.
Does the propranolol, alone or in combination with other chemotherapy drugs, ever been tested on cancer stem cells?
The references used in the manuscript, although not all recent, were adequate to support the evidence.
Spelling errors were detected.
Minor points:
Line 65: The authors should add the references of the cited works.
Line 271: Please correct “pre -and” in “pre- and”
Author Response
Rebuttal letter: manuscript # jcm-1799453.
Propranolol: A “Pick and Roll” Team Player in Benign Tumors and Cancer Therapies, as a contribution to the Special Issue: Present and Future of Targeted Molecular Therapies in Solid Tumors – Clinician’s Perspective.
Dear Editor and Reviewers,
We thank the editor and the four reviewers for the time they spent looking over the manuscript. We also appreciate the comments and suggestions done to improve quality and the clarity of the manuscript.
Below is our response to each point raised by the reviewers.
All major changes made to the original manuscript have been highlighted in red. Furthermore, the English of the text has been edited by a native speaker to improve the quality of it.
We have also modified tables following the reviewers’ comments.
We ask for the Editor in charge decision about an opposite point of view between one of the reviewers and the authors.
Finally, we hope that we satisfyingly addressed them all and that the manuscript will be now suited for publication.
Sincerely,
On behalf of all authors,
Luisa María Botella and Angel M Cuesta.
Reviewer 4:
Comments and Suggestions for Authors:
Within their manuscript, the authors give an interesting overview about actual information about propanolol with a focus on its clinical application in cancer treatment. In addition, they report all the clinical trials on propranolol use. The information in the manuscript can be helpful in recapitulating some aspects, pointing in specific critical points.
The literature research did not show any similar review like this one, so any plagiarism was detected.
The structure and layout of the manuscript were written in an organized way, easy to read and understand; however, sometime, there are no references of in vitro studies evaluating the effect of propanolol on cancer cells. I fully understand that this Journal call for the clinical topic but to improve the quality of the manuscript, I believe should be important insert some works rely on the effectiveness of propanolol on cancer cells.
We appreciate the suggestion, in fact, during the design of this review, we have also been discussing whether it is adequate to include “the origin of propranolol” in a Special Issue named: "Present and Future of Targeted Molecular Therapies in Solid Tumors – Clinician’s Perspective".
In this sense, we have included (and highlighted in red), in section 2, new paragraphs describing a Historical perspective of propranolol. We hope it fits with what the reviewer had in mind. The text is as follows:
“More than a century ago scientist speculated that catecholamines, after binding selectively to receptor-like structures, caused their pharmacological actions. Was in 1948 when R. P. Ahlquist described, for the first time, the α-and β-adrenotropic receptors (PMID: 18882199). Some years later, Sir James Whyte Black, a Scottish pharmacologist at the Imperial Chemical Industries (ICI) in UK, was searching for a chemically synthesized agent to interfere catecholamines and, therefore, decrease the oxygen heart requirements and relieve the pain of angina. He was searching for a β-adrenergic blocker.
There were two main candidates: pronethalol and propranolol (also named ICI 45,520 or, since 1964, marketed as Inderal). The latter is a more potent derivative from the former and lacks the carcinogenicity founded in animal models treated with pronethalol. Then, propranolol became a best-selling drug, used to treat a wide range of cardiovascular diseases such as arrhythmia, hypertension, and hypertrophic cardiomyopathy. Finally, in 1988, Sir James W. Black was awarded the Nobel Prize in Medicine for creation of propranolol (PMID: 9456487).
Nowadays, propranolol has fallen from the 1st line treatment since it does not perform as well as other drugs. Nevertheless, evidence began to accumulate paving the potential use (reposition) of propranolol as an antitumor drug. This use is supported by several case reports [9–12] that postulated the use of propranolol in a rare tumor like pheochromocytoma to reduce the hypertension generated by catecholamines secretion. Interestingly, these studies outlined (but did not relate) the absence of metastases and tumor growth amelioration without noticeable side effects, together with normalization of heart rate and blood pressure. Few years later, different reports started to show clinical benefits of propranolol in different hyperproliferative diseases such as astrocytomas (PMID: 4330940), epidermoid carcinoma (PMID: 165357 and PMID: 6143691), malignant insulinoma (PMID: 6302161), lung adenocarcinoma (PMID: 2569945), and breast cancer (PMID: 10845278).
Nevertheless, propranolol continued on the bench as cardioprotective medicine until…”
Does the propranolol, alone or in combination with other chemotherapy drugs, ever been tested on cancer stem cells?
We thank the very important question about stem cells. To the best of our knowledge, we have found a few of articles and all but one is focused on a benign tumor such as hemangioma (PMID: 26574555; PMID: 24427325 and PMID: 23096601). Zhang et al., (PMID: 24427325), the other article speaks about vascular embryonic stem cells (PMID: 25130141).
One of the points for this review is to put light in the increment of clinical trials using propranolol as single treatment or as adjuvant with other therapies like as surgery or other chemotherapeutics. We consider that this is based on the increment of promising results in basic cancer research and this, as a natural consequence/flow will end, sooner or later, in studies in cancer stem cells.
The references used in the manuscript, although not all recent, were adequate to support the evidence.
We agree with the comment, but the purpose of these references is to show the initial data on which recent studies and trials are based.
Spelling errors were detected.
We have revised the text and fixed the spelling errors.
Minor points:
Line 65: The authors should add the references of the cited works.
Thank you for this important suggestion. We have modified the sentence (and highlighted in red) as follows: “In addition, accumulated publications and clinical trials point to its possible role as a regulator of the tumor microenvironment (see section 2. Why propranolol?). As a paradigmatic example, propranolol has become the Most Valuable Player drug in the benign tumor Infantile Hemangioma (IH) (see Supp Table 1)”.
Line 271: Please correct “pre -and” in “pre- and”
Thank you, this has been fixed.
Finally, we again appreciate all the valuable comments from reviewers that have led us to rewrite and improve the manuscript in many aspects.
REFERENCES
PMID: 10845278. Slotkin TA, Zhang J, Dancel R, Garcia SJ, Willis C, Seidler FJ. Beta-adrenoceptor signaling and its control of cell replication in MDA-MB-231 human breast cancer cells. Breast Cancer Res Treat. 2000 Mar;60(2):153-66. doi: 10.1023/a:1006338232150.
PMID: 12418927. Sommers Smith SK, Smith DM. Beta blockade induces apoptosis in cultured capillary endothelial cells. In Vitro Cell Dev Biol Anim. 2002 May;38(5):298-304. doi: 10.1290/1071-2690(2002)038<0298:BBIAIC>2.0.CO;2.
PMID: 165357. Kelly LA, Butcher RW. Studies on cyclic AMP metabolism in human epidermoid carcinoma (HEp-2) cells. Metabolism. 1975 Mar;24(3):359-68. doi: 10.1016/0026-0495(75)90116-x.
PMID: 18882199. AHLQUIST RP. A study of the adrenotropic receptors. Am J Physiol. 1948 Jun;153(3):586-600. doi: 10.1152/ajplegacy.1948.153.3.586.
PMID: 19467330. Annabi B, Lachambre MP, Plouffe K, Moumdjian R, Béliveau R. Propranolol adrenergic blockade inhibits human brain endothelial cells tubulogenesis and matrix metalloproteinase-9 secretion. Pharmacol Res. 2009 Nov;60(5):438-45. doi: 10.1016/j.phrs.2009.05.005. Epub 2009 May 23.
PMID: 19946348. Hajighasemi F, Hajighasemi S. Effect of propranolol on angiogenic factors in human hematopoietic cell lines in vitro. Iran Biomed J. 2009 Oct;13(4):223-8.
PMID: 20732454. Lamy S, Lachambre MP, Lord-Dufour S, Béliveau R. Propranolol suppresses angiogenesis in vitro: inhibition of proliferation, migration, and differentiation of endothelial cells. Vascul Pharmacol. 2010 Nov-Dec;53(5-6):200-8. doi: 10.1016/j.vph.2010.08.002. Epub 2010 Aug 20.
PMID: 20739470. Ristori C, Filippi L, Dal Monte M, Martini D, Cammalleri M, Fortunato P, la Marca G, Fiorini P, Bagnoli P. Role of the adrenergic system in a mouse model of oxygen-induced retinopathy: antiangiogenic effects of beta-adrenoreceptor blockade. Invest Ophthalmol Vis Sci. 2011 Jan 5;52(1):155-70. doi: 10.1167/iovs.10-5536.
PMID: 22580939. Chim H, Armijo BS, Miller E, Gliniak C, Serret MA, Gosain AK. Propranolol induces regression of hemangioma cells through HIF-1α-mediated inhibition of VEGF-A. Ann Surg. 2012 Jul;256(1):146-56. doi: 10.1097/SLA.0b013e318254ce7a.
PMID: 22743651. Chisholm KM, Chang KW, Truong MT, Kwok S, West RB, Heerema-McKenney AE. β-Adrenergic receptor expression in vascular tumors. Mod Pathol. 2012 Nov;25(11):1446-51. doi: 10.1038/modpathol.2012.108. Epub 2012 Jun 29.
PMID: 23096601. Wong A, Hardy KL, Kitajewski AM, Shawber CJ, Kitajewski JK, Wu JK. Propranolol accelerates adipogenesis in hemangioma stem cells and causes apoptosis of hemangioma endothelial cells. Plast Reconstr Surg. 2012 Nov;130(5):1012-1021. doi: 10.1097/PRS.0b013e318267d3db.
PMID: 23695022. Pasquier E, Street J, Pouchy C, Carre M, Gifford AJ, Murray J, Norris MD, Trahair T, Andre N, Kavallaris M. β-blockers increase response to chemotherapy via direct antitumour and anti-angiogenic mechanisms in neuroblastoma. Br J Cancer. 2013 Jun 25;108(12):2485-94. doi: 10.1038/bjc.2013.205. Epub 2013 May 21.
PMID: 24427325. Zhang L, Mai HM, Zheng J, Zheng JW, Wang YA, Qin ZP, Li KL. Propranolol inhibits angiogenesis via down-regulating the expression of vascular endothelial growth factor in hemangioma derived stem cell. Int J Clin Exp Pathol. 2013 Dec 15;7(1):48-55.
PMID: 25130141. Sharifpanah F, Saliu F, Bekhite MM, Wartenberg M, Sauer H. β-Adrenergic receptor antagonists inhibit vasculogenesis of embryonic stem cells by downregulation of nitric oxide generation and interference with VEGF signalling. Cell Tissue Res. 2014 Nov;358(2):443-52. doi: 10.1007/s00441-014-1976-8. Epub 2014 Aug 19.
PMID: 2569945: Schuller, H.M.; Cole, B. Regulation of cell proliferation by β-adrenergjc receptors in a human lung adenocarcinoma cell line. Carcinogenesis 1989, 10, 1753-1755.
PMID: 25728347. Pan WK, Li P, Guo ZT, Huang Q, Gao Y. Propranolol induces regression of hemangioma cells via the down-regulation of the PI3K/Akt/eNOS/VEGF pathway. Pediatr Blood Cancer. 2015 Aug;62(8):1414-20. doi: 10.1002/pbc.25453. Epub 2015 Mar 1.
PMID: 26394686. Albiñana, V.; Villar Gómez De Las Heras, K.; Serrano-Heras, G.; Segura, T.; Perona-Moratalla, A.B.; Mota-Pérez, M.; de Campos, J.M.; Botella, L.M. Propranolol Reduces Viability and Induces Apoptosis in Hemangioblastoma Cells from von Hippel-Lindau Patients. Orphanet J Rare Dis 2015, 10, 118.
PMID: 26574555. Munabi, N.C.; England, R.W., Edwards, A.K., Kitajewski, A.A.; Tan, Q.K.; Weinstein, A.; Kung, J.E.; Wilcox, M.; Kitajewski, J.K.; Shawber, C.J.; Wu, J.K. Propranolol Targets Hemangioma Stem Cells via cAMP and Mitogen-Activated Protein Kinase Regulation. Stem Cells Transl Med 2016, 5, 45-55.
PMID: 27085011. Khouri C, Jouve T, Blaise S, Carpentier P, Cracowski JL, Roustit M. Peripheral vasoconstriction induced by β-adrenoceptor blockers: a systematic review and a network meta-analysis. Br J Clin Pharmacol. 2016 Aug;82(2):549-60. doi: 10.1111/bcp.12980. Epub 2016 May 31.
PMID: 27906047. Cardwell CR, Pottegård A, Vaes E, Garmo H, Murray LJ, Brown C, Vissers PA, O'Rorke M, Visvanathan K, Cronin-Fenton D, De Schutter H, Lambe M, Powe DG, van Herk-Sukel MP, Gavin A, Friis S, Sharp L, Bennett K. Propranolol and survival from breast cancer: a pooled analysis of European breast cancer cohorts. Breast Cancer Res. 2016 Dec 1;18(1):119. doi: 10.1186/s13058-016-0782-5.
PMID: 28031536. Montoya A, Amaya CN, Belmont A, Diab N, Trevino R, Villanueva G, Rains S, Sanchez LA, Badri N, Otoukesh S, Khammanivong A, Liss D, Baca ST, Aguilera RJ, Dickerson EB, Torabi A, Dwivedi AK, Abbas A, Chambers K, Bryan BA, Nahleh Z. Use of non-selective β-blockers is associated with decreased tumor proliferative indices in early stage breast cancer. Oncotarget. 2017 Jan 24;8(4):6446-6460. doi: 10.18632/oncotarget.14119.
PMID: 31296894. Cuesta, A.M.; Albiñana, V.; Gallardo-Vara, E.; Recio-Poveda, L.; de Rojas-P, I.; de Las Heras, K.V.G.; Aguirre, D.T.; Botella, L.M. The Β2-Adrenergic Receptor Antagonist ICI-118,551 Blocks the Constitutively Activated HIF Signalling in Hemangioblastomas from von Hippel-Lindau Disease. Sci Rep 2019, 9, 10062.
PMID: 31607128: Spini, A.; Roberto, G.; Gini, R.; Bartolini, C.; Bazzani, L.; Donnini, S.; Crispino, S.; Ziche, M. Evidence of β-blockers drug repurposing for the treatment of triple negative breast cancer: A systematic review. Neoplasma 2019, 66, 963-970.
PMID: 35784377. Wilson S, Hassan D, Jakeman M, Breuning E. Cost-effectiveness of atenolol compared to propranolol as first-line treatment of infantile haemangioma: A pilot study. JPRAS Open. 2022 May 26;33:52-56. doi: 10.1016/j.jpra.2022.05.010.
PMID: 4330940. Clark RB, Perkins JP. Regulation of adenosine 3':5'-cyclic monophosphate concentration in cultured human astrocytoma cells by catecholamines and histamine. Proc Natl Acad Sci U S A. 1971 Nov;68(11):2757-60. doi: 10.1073/pnas.68.11.2757.
PMID: 6143691. Delavier-Klutchko C, Hoebeke J, Strosberg AD. The human carcinoma cell line A431 possesses large numbers of functional beta-adrenergic receptors. FEBS Lett. 1984 Apr 24;169(2):151-5. doi: 10.1016/0014-5793(84)80308-7.
PMID: 6302161. Blum I, Rusecki Y, Doron M, Lahav M, Laron Z, Atsmon A. Evidence for a therapeutic effect of dl-propranolol in benign and malignant insulinoma: report of three cases. J Endocrinol Invest. 1983 Feb;6(1):41-5. doi: 10.1007/BF03350559.
PMID: 9456487. Stapleton MP. Sir James Black and propranolol. The role of the basic sciences in the history of cardiovascular pharmacology. Tex Heart Inst J. 1997;24(4):336-42.

Round 2
Reviewer 2 Report
Dear authors,
Thank you for reply carefully to my comments.
Author Response
Dear reviewer and Editor.
Here I send the latest version of the manuscript, including the changes done in red.
We thank again the tme and effort dedicated to our manuscript.
Kind Regards
The authors
